# Water and Climate Change, Two Key Objectives in the Agenda 2030: Assessment of Climate Literacy Levels and Social Representations in Academics from Three Climate Contexts

**Amor Escoz-Roldán** [1] , **José Gutiérrez-Pérez** [1,*] **and Pablo Ángel Meira-Cartea** [2]

1   Department of Educational Methodology Research, Universidad de Granada, 18071 Granada, Spain;
    amorescoz@correo.ugr.es
2   Department Pedagogy and Didactic, Universidad de Santiago de Compostela,
    15705 Santiago de Compostela, Spain; pablo.meira@usc.es
*   Correspondence: jguti@ugr.es

**Abstract:** The relationship between climate change and water is an obvious and key issue within the United Nations Sustainable Development Goals. This study aims to investigate the social representation created around this relationship in three different territorial contexts in order to evaluate the influence of the territory on the perception of the risk of climate change and its relationship with water. By means of a questionnaire completed by 1709 university students, the climatic literacy of the individual was evaluated in order to relate it to other dimensions on the relationship between climate change and water (information, training previous on climate change and pro-environmental attitudes) in their different dimensions in three different territorial contexts. Three hypotheses have been tested: (1) The denial of the CC is significantly associated with a representation that belittles the consequences of global warming and other extreme phenomena. (2) Territorial contexts with high average rainfall levels and low average annual temperatures tend to minimize the social representation of water risks associated with the CC. (3) There is significant interaction between the socio-cultural context and social representations on the causes, consequences and solutions to the problems of CC and water. The first two hypotheses have been rejected, while the third has been accepted. The research results show high climate literacy in the samples of selected university students. It is noted that students recognize a close relationship between the problem of water and the climate crisis. Likewise, they identify different types of causes, consequences, physical processes and solutions. Different climatological contexts do not show significant differences in the social representations that students show about climate change, while socio-educational variables such as available scientific information, or ideology orientation do show significant differences.

**Keywords:** water; climate change; territorial context; Sustainable Development Goals; Agenda 2030; university students; climate literacy; social representation

## 1. Introduction

Water, that simple molecule composed of two hydrogen atoms and one oxygen atom, is one of the essential elements of the planet. Without water there is no life. Paradoxically, however, this particularly important fact does not seem to concern us, since until a decade ago water supply and sanitation was not considered an expressly recognized human right [1]: "All persons have the right to sufficient, safe, physically accessible, affordable and of acceptable quality water for personal and domestic use on a continuous basis".

Water is a basic necessity for all living beings, essential in the configuration of environmental systems. It constitutes more than 80% of the body of most living beings, intervenes in their metabolic processes and is a fundamental part of the photosynthesis of plants, in addition to being the habitat of a wide variety of living beings. We depend on water to generate and maintain activities such as agriculture, fishing, energy production, industry, transport or tourism. Depending on their availability, we decide where to settle our population centres and how to occupy the territory; being a source of geopolitical conflicts when it is scarce [2].

According to the World Health Organization, some 5200 million people use safe and uncontaminated managed drinking water services, while nearly 30% of other people do not have direct drinking water services, so more than 1300 billion have access to an improved water source within 30 minutes (on a round trip), 263 million have an improved water source more than 30 minutes away, 423 million people draw water from unprotected wells and springs, and 159 million people collect untreated surface water from lakes, ponds, rivers or streams. Poor sanitation and pollution lead to the transmission of diseases such as cholera, dysentery, hepatitis, typhoid fever, polio, trachoma, intestinal worms, dengue or schistosomiasi, ... through exposure to infested water. Lack of water, sanitation and hygiene are the main causes of neglected tropical diseases that affect more than 1500 million people each year [3].

When water comes from safe sources, this translates into positive economic and social consequences and a significant reduction in human disease and risk. However, water as a natural resource and an essential element for life and ecosystems on the planet is being severely affected by the consequences of climate change (CC). New scenarios of crisis and risk derived from the increase of the temperatures, the increase of the levels of evaporation, the torrential rains, floods, thaw, rise of the sea level, advance of the deforestation and desertification... force the investigators to dedicate more attention to this topics, focusing the problem from the complexity of their interactions, and not as isolated and independent aspects. The scarcity of research undertaken in the field of social sciences [4] justifies the study we present. The response of human societies to the climate crisis will not depend solely on the best available science. The 'human factor', in all its dimensions and expressions, is going to be fundamental in avoiding the worst-case scenarios of the future. It is necessary to increase the contribution of the Social Sciences to the knowledge of how people and human communities interpret, value, act and change—or not—in the face of the climate crisis. The incorporation of the social scientists' perspective into the politics on the CC is feasible and necessary [5].

Working Group III of the IPCC has demanded socio-environmental research beyond the hard science agenda. The report identifies more than 20 topics for future research along these lines, including the fields of study of behavioural sciences, education and communication [6]; it also underlines the need to focus social research on reducing epistemic uncertainty about social perceptions and responses to the CC.

There is significant evidence in the body of empirical literature concerning the influence of certain variables on the recognition of the CC as an established scientific phenomenon on which there is consensus in the research community [7]. Recently, psychosocial and educational aspects such as the influence of the perception of the seriousness of environmental problems, personal experiences in this regard, the proximity or remoteness of finished specific problems, the effectiveness of certain educational programs have begun to gain interest. In this study we focus on an evaluation of the social representations of citizens of southern Europe, who reside in three territorial contexts of the Iberian Peninsula, with different climatic and cultural conditions. We try to reveal the incidence of these factors, together with a series of other context variables (climate denialism, academic cultures, perception of the severity of risks, involvement in pro-environmental activities and associations, among others) and their influence on literacy on CC and its interaction with different dimensions related to water, its natural cycle, decisions on its management and sanitation, as well as the consequences for populations and ecosystems.

## 2. Water Crisis, CC and Objectives of Research

*2.1. Water Crisis and CC*

The water crisis can be defined in different contexts, related to scarcity, insecurity, availability, unhealthiness, potabilization, demand, the threat of floods; contexts intimately related to each other and directly associated with the consequences of the CC:

(1)　First, we talk about scarcity and insecurity, as water availability per person worldwide has fallen by 55% since 1960, and demand is expected to grow by 50% by 2030 [8]. Considering the above population increase (9.7 million in 2050), it is estimated that, of these, 3900 million will live in river basins extremely affected by water stress, which means that the forecast for 2050 is that water demand will increase by 400% for industry and 130% for domestic use [9]. Human insecurity linked to water is exacerbated by drought, affecting more people than any other type of natural disaster. One example is the 411 million people affected by natural disasters in 2016, 94% of whom were caused by drought, with direct consequences for agriculture. There are also areas on the planet where water stress is greater due to the increase in the economy and population, which requires more land for food production and, therefore, greater difficulties in managing water resources properly and sustainably, which will be even more difficult due to the CC (temperature increases that will lead to greater evaporation of water). If in addition to this transboundary conflicts are added by the management of the water of a river by different countries with different interests, the problem is aggravated even more, making the availability of water less due to pollution, construction of dams, population demand, etc., which makes it very necessary to propose models of future socioeconomic scenarios to adapt sustainably to the increase in water demand and ensure food security which is intimately related to water and climate [10,11].

(2)　Water-related disasters account for 70% of all deaths related to extreme weather events [12]. It is estimated that by 2050 between 150 and 200 million people may have to leave their areas of residence due to phenomena such as desertification, the increase in extreme weather events—floods have been the most frequent global natural disaster in the last two decades [13] or rising sea levels [14]. In addition, the population living on land prone to flooding, the consequences of climate change, deforestation, loss of wetlands and rising sea levels are expected to increase this year, increasing the number of people vulnerable to flood disasters by 2 billion [15].

(3)　A third context in the water crisis is sanitation and its relationship to health. Although improvements in supply have been increasing, 663 million people did not have access to improved drinking water sources in 2015 [16]. Even so, these sources are not always safe: according to WHO, some 45 million people in Bangladesh drink water with arsenic concentrations higher than those permitted by WHO. On the other hand, when it comes to women and girls, sanitation services are even more important because they are intimately related to their health, which is put at risk when they are absent or unsafe. In addition, in the case of children, diarrhoeal diseases caused by poor sanitation cause one in nine deaths, with diarrhoea being the third leading cause of death in children under five worldwide despites being an easily preventable infection. In a 2015 survey of low- and middle-income countries, 38 per cent of health facilities did not have access to a source of safe drinking water, 35% had neither soap nor water, and 19% had no improved sanitation, exacerbating the problem [9].

(4)　Another determinant of the water crisis is the current pace of development, as not enough is being invested in water supply, sanitation and hygiene: in order to achieve the water-related Sustainable Development Goals (SDG), three times more capital would be needed than the current investment [17]. The rampant increase in meat consumption is causing the consumption of water for livestock to soar: while to produce 1 kg of rice requires 3500 litres of water, to produce 1 kg of meat requires 15,000 litres, adding that methane emissions from livestock wastewater could increase by 50% and nitrous oxide emissions by 25% between 1990 and 2020 [18]. Overall, the food

industry in both low- and high-income countries contributes 54% and 40%, respectively, to the discharge of organic pollutants into water [19]. On the other hand, other types of human activities also degrade water resources; without going any further, two million tons of human waste are emptied into watercourses every day [20] and an estimated 15–18 million $m^3$ of freshwater resources are polluted by fossil fuels [21].

(5) Ecosystem degradation is another expression of the global water crisis. The 12.6 million global deaths attributed to the environment in 2012 [22] are clear evidence that environmental degradation is intimately linked to health. By 2050, the number of eutrophicated lakes is expected to increase by 20%, which means that by the same date one third of the world's population will face risks from excess nitrogen and phosphorus in water associated with this phenomenon [23]. On the other hand, since 1900, 64% of the world's wetlands have disappeared [24] and it is estimated that in the period 1970–2010 populations of freshwater species declined by about 76% [25]. In addition, one third of the world's amphibians are at risk of extinction, as are 50% of native freshwater fish species [26].

(6) Another very important example of ecosystem degradation is the alteration of peatlands. Although they cover only 3% of the Earth's surface, if they remain humid they can store approximately twice as much carbon as all the world's forests combined. The loss of 15% of these ecosystems would cause a contribution equal to 5% of anthropogenic $CO_2$ emissions worldwide [27]. In the Nordic and Baltic states, 45% of peatlands have been drained, which is currently emitting approximately 80 megatons of $CO_2$ per year, accounting for 25% of these countries' total $CO_2$ emissions.

(7) Among the most degraded ecosystems are rivers and oceans. Due to the enormous amount of plastics, among other waste, that we dispose of in them. According to a report by the European Environment Agency, it is estimated that each year 10 million tonnes of waste are dumped into the sea, plastics being the most common type of waste because of the exponential increase in the production of these materials since 1950, going from 1.5 million tonnes per year to 280 million tonnes today. Of the 10 million tons of garbage that end up in the oceans, 8 million tons are plastics; a quarter of this amount comes from only ten rivers in the world and eight of those rivers are in Asia. The researchers, through a model that included data from studies on 57 rivers in different parts of the world, found that they pour between half a million and 2.75 million tons of plastic into the sea each year and the ten that transport 93% of these plastics are the rivers Yangtsé, Amarillo, Hai, de las Perlas, Amur, Mekong, Indo and Ganges in Asia, and the Niger and Nile rivers in Africa. The Yangtze River alone discharges up to 1.5 million tons of plastic waste annually into the Yellow Sea [28].

(8) In addition to providing a high value ecosystem service, water is an indispensable element for the life of all living organisms on the planet and is also a vector for climate and weather regulation. The flow of clean, uncontaminated water ensures the sustainability of ecosystems and increases the likelihood of people's health. This resource is not unlimited as one would expect from the perception that one often has of its cycle. That is why knowing the perception of water in all its spheres (health, hygiene, development, climate regulation, etc.) becomes especially important to act in a more incisive way through education and awareness, both in children and adults.

(9) Coastal flood hazard modelling scenarios from sea-level rise by 2050 estimate that about 300 million people live in flood-prone coastal areas. The highest risk areas of the Iberian Peninsula are located in Doñana, Delta del Ebro, Manga del Mar Menor and coastal municipalities of Huelva and Cadiz [29].

## 2.2. Objectives of Research

- $O_{\#1}$: To assess the degree of climate literacy around the relationship between water and climate change (extreme weather events, rising sea levels, desertification, etc.) in university students from three different climatological and cultural contexts of the Iberian Peninsula.

- O$_{\#2}$: To assess whether the climate literacy of these students corresponds to pro-environmental attitudes and the information they claim to have on different aspects of the CC and its relationship with water.

- O$_{\#3}$: To compare the results in the three contexts analysed in order to determine which factor (territory, climate or common culture) influences the social representation of university students around the relationship between water and the CC.

As indicated in the previous points, and in order to better understand the objectives of this research, is presented below the specific problem suffered by the Iberian Peninsula in relation to the water crisis due, among other aspects, to CC.

## 3. Background

According to WWF's October 2019 *Water Scarcity and Droughts Report* [30], the Iberian Peninsula has traditionally lived with scarce and highly variable water resources and will have to face increasingly severe extreme phenomena in the near future. Droughts are natural and recurrent phenomena in the Iberian Peninsula due to its predominantly Mediterranean climate, with a very variable rainfall regime and, on the other hand, water scarcity problems arise once water demand and supply are unbalanced.

In the Iberian Peninsula, drought episodes have increased in duration and severity, and the uncertainties to prevent them are very high. As the water regime is very variable and with a marked dry season, the Iberian Peninsula has high variability of annual rainfall, and because of these conditions, most rivers are temporary and wetlands are fully adapted to suffer low water levels and even dry completely for many months as part of their ecological requirements. Furthermore, rivers depend to a large extent on their interactions with aquifers when they are connected, which is part of the natural response to the annual dry season and eventual droughts, which guarantees the health of the aquatic ecosystems of the Iberian Peninsula both of rivers and of wetlands and aquifers, and constitutes the basis of their state of conservation.

However, both Portugal and Spain have a very high demand for water for different uses related to an unsustainable increase in intensive agriculture that has led to the modification and regulation with large dams of the vast majority of the rivers flowing in the Iberian Peninsula, in order to supply water to irrigators, which has led to the drying of much of the wetlands of both countries, in order to recover fertile land for agriculture.

On the other hand, changes in land use and vegetation, due to urbanization and the expansion of intensive agriculture have significantly increased the risk of desertification and aridity in many areas characterized by high temperatures and low rainfall. In addition, in large parts of Spain and Portugal the natural and adapted characteristics of typical aquatic ecosystems have been destroyed to cope with dry seasons and periods of drought and many of the aquifers suffer one of the highest exploitation rates in Europe, which poses an additional threat to these "natural reserves" for aquatic ecosystems during these dry periods.

### 3.1. Territorial Contextualisation of the Study

The territorial areas chosen for this study are three cities: Granada, Santiago de Compostela and Braga. All three are located in the Iberian Peninsula, which is located in southwestern Europe surrounded by the Mediterranean Sea and the Atlantic Ocean, joining the rest of the continent in the northeast.

Almost the entire surface of the peninsula is occupied by Spain and Portugal. The peninsula is 582,918 km$^2$, of which 493,515 km$^2$ belong to Spain, 88,944 km$^2$ to Portugal, 453 km$^2$ to Andorra and 6 km$^2$ to Gibraltar. For this study we will only consider Spain and Portugal. The geographical uniqueness of the Iberian Peninsula is due to its location and configuration since it is located in the Mediterranean area, in the extreme southwest of the European continent, between two continents (Europe and Africa) and between two seas (Atlantic and Mediterranean) [31].



To the south, the peninsula is separated from Africa by the Mediterranean Sea, an area known as the Alborán Sea, and the Atlantic Ocean, the Strait of Gibraltar being the boundary between them. The highest point is the Mulhacén (Sierra Nevada, Granada) of 3478.6 m above sea level. The longest river is the Tagus, with a length of 1007 km, of which 731 km are in Spain and 275 km in Portugal.

In general terms, the most widespread citizen perception of the Iberian Peninsula is that of a dry territory, with the exception of the northernmost regions. This social representation is a kind of empirical axiom, a truth that does not need to be demonstrated in the light of the landscape evidence: "one sees, lives, enters through the eyes... However, more than a scientific truth, it is an empirical perception, an experience based on two fundamental facts: the dryness of summer and the frequent irregularity of rainfall during the rest of the year on the one hand and, on the other, the desolate, dry, sub-desert visual landscape that our territory often offers" [32]. In strictly scientific terms, the reality is different. It is true that it rains little or rains less than it can evaporate, which led classical studies to identify a dry Spain (with a negative global water balance, with evapotranspiration exceeding precipitation levels) and a wet Spain (with a positive global balance) [33]. This view is also reductionist, since the Iberian Peninsula presents a great variety of climates due to its geographical position and orography. Being located at the southern limit of influence of the polar front, with its associated squalls, it presents features of the humid continental climate of the western part of Europe. In addition, being in the northern limit of action of the zones of high tropical pressures, which carry warm and dry air, there are also climatic rests associated with the desert areas of Saharan Africa.

The north of the peninsula is more influenced by the cyclonic system of squalls, while the south is dominated by a more tropical climate. Due to the dynamics of the atmosphere, it is frequent that during the winter the humid fronts coming from the Atlantic sweep the peninsula, provoking intense rains.

In summer, influenced by high tropical pressures, the Azores anticyclone intensifies, leading to hot and dry weather that has little effect on the Cantabrian Coast, which is more influenced by the Atlantic fronts, although to a lesser extent during the summer period. It can be said that the annual climatic cycle of the peninsula has two main seasons, summer and winter, as both spring and autumn are transition seasons.

The climates of the Iberian Peninsula are conditioned by its abrupt relief, characterized by numerous mountainous systems that concentrate mainly in its periphery, isolating it from the marine influence except for the western zone. In this way, when the fronts of rains of the Atlantic penetrate in the peninsula, they cross it unloading the water until colliding with some of the mountainous systems, not being able to surpass them and creating areas of pluviometric shade, where the precipitation is smaller than in nearby places. This situation favours the appearance of arid territories in the south eastern part of the peninsula, as well as in other inland regions. In other cases, the local orography has the opposite effect, when the downwind slopes of the mountains collect all the rain carried by the fronts, increasing precipitation in certain areas.

The Iberian Peninsula (Figure 1) can thus be said to be divided into three large zones on the basis of the Köppen classification (the Köppen system is based on the fact that natural vegetation has a clear relationship with climate, so the boundaries between one climate and another were established taking into account the distribution of vegetation. The parameters for determining the climate of an area are the average annual and monthly temperatures and rainfall, and the seasonality of the precipitation. It divides the world's climates into five main groups: tropical, dry, temperate, continental and polar, identified by the first letter in capital letters. Each group is divided into subgroups, and each subgroup into climate types. Climate types are identified with a 2- or 3-letter symbol [34]). The first has a semi-arid Mediterranean climate, i.e. steppe, with a semi-arid south-eastern zone, transition between the steppe and the desert.

The second zone occupies a narrow coastal strip that begins between the mouths of the Tagus and Duero rivers, rises to the north and runs along the entire Cantabrian Cornice. Its climate would be of the maritime type of the west coast, with regions of sub humid subtype and others of humid subtype. The third region is smaller than the previous one, starting at the western end of the Cantabrian

Mountains and ending at the eastern end of the Pyrenees. It is characterized by a climate typical of areas located at high altitudes. In this study we focus on the first two areas. On the other hand, the entire Levante and the southern half of the peninsula correspond to a temperate climate with dry and hot summers.

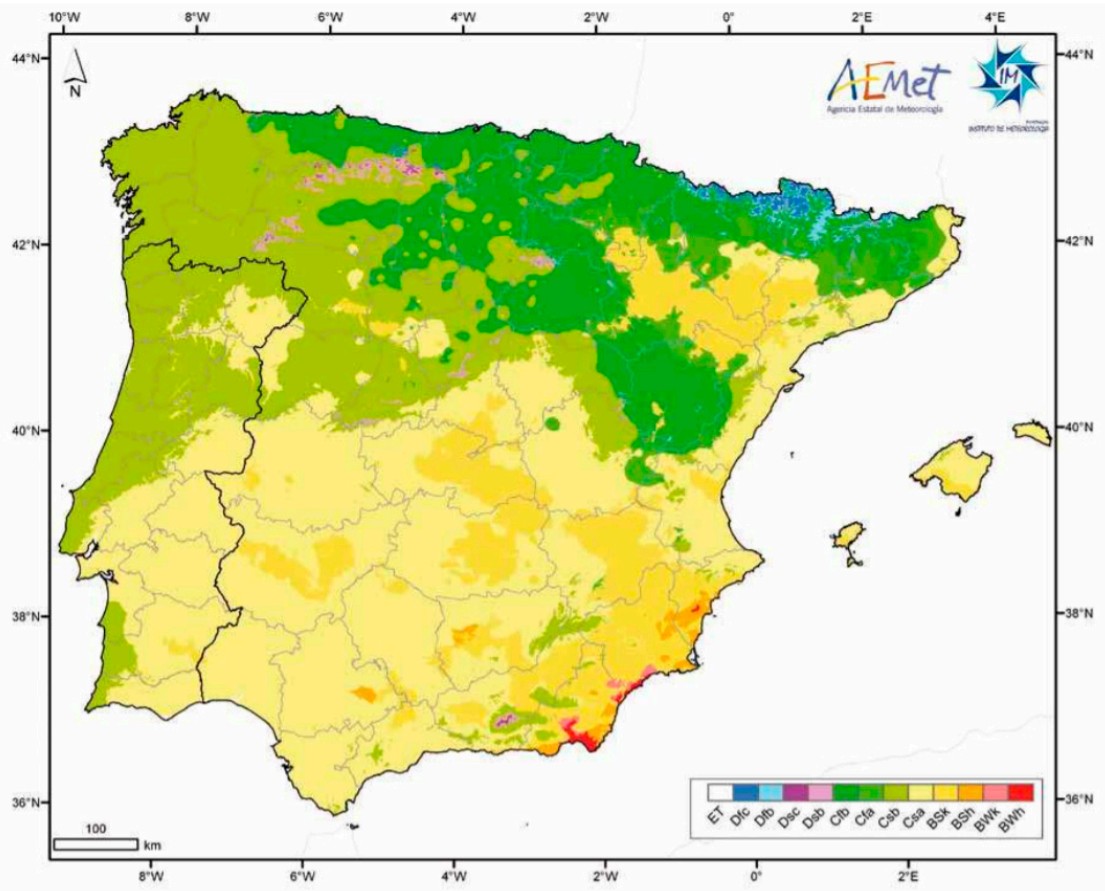

**Figure 1.** Köppen-Geiger Classification of the Iberian Peninsula and Balearic Islands [35].

The Atlantic zone occupies the peninsular regions in contact with the Atlantic Ocean, from which they receive a great influence and moderates their temperatures; directly affected by the fronts that come from the sea that give it a more humid climate. This area occupies the north of the peninsula, from the eastern end of the Pyrenees to Galicia, the west, covering the western strip of Portugal and much of the coastal areas of Andalusia to the east of the province of Granada.

Another zone of Atlantic influence is found in the interior of Portugal, where the oceanic influence is still high but as it penetrates into the interior there are continental features of the climate that make it more extreme, with reduced rainfall and increasing average temperatures.

The classic continental climate is located in both plateaus, in the Ebro valley and in areas of the eastern interior of Andalusia, with hot summers and cold winters. Rainfall is scarce, giving rise essentially to a climate that could be classified as semi-arid [36]. The following climogram Figures 2–4 and Tables 1–3) summarise the climatological characteristics of each of the three territorial contexts of the study:

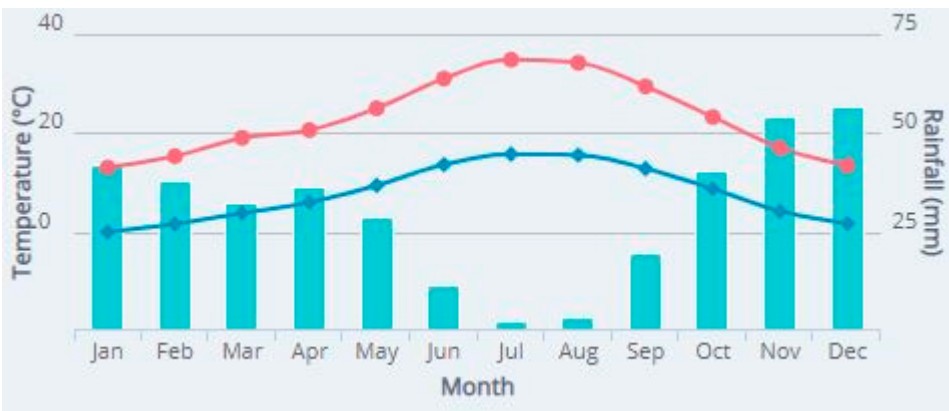

**Figure 2.** Climatological information based on monthly averages for the 30-year period 1971–2000 of the Territorial Context 1 [37].

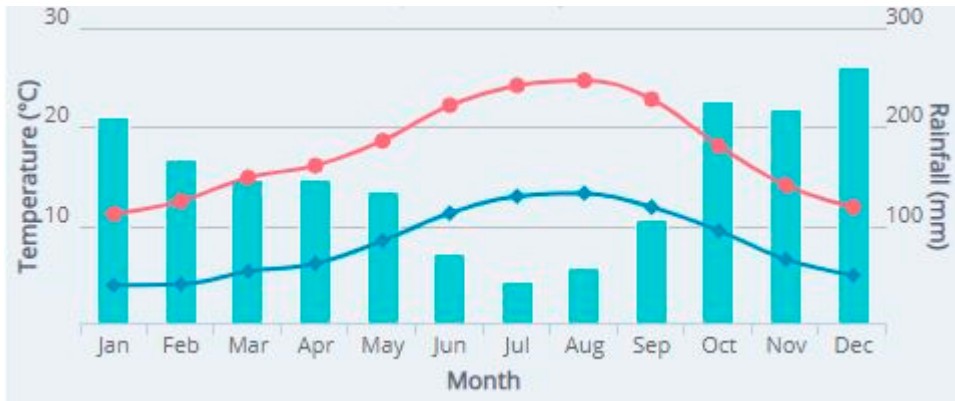

**Figure 3.** Climatological information based on monthly averages for the 30-year period 1971–2000 of the Territorial Context 2 [38].

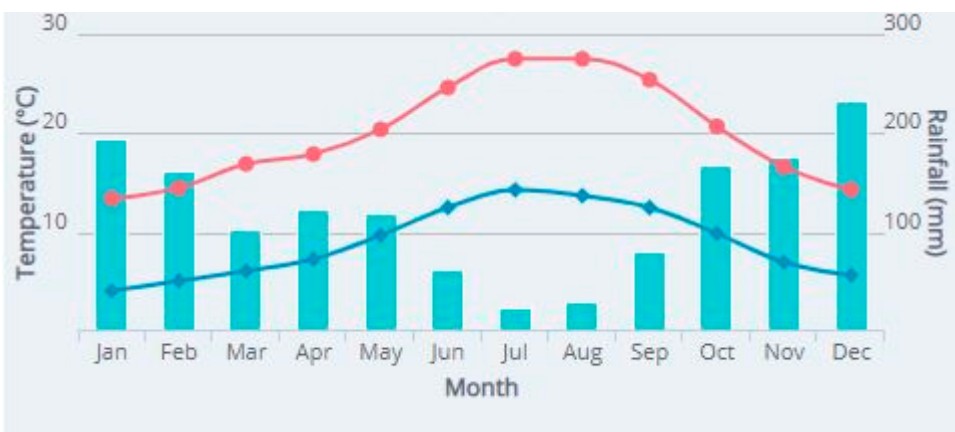

**Figure 4.** Climatological information based on monthly averages for the 30-year period 1971–2000 of the Territorial Context 3 [39].

**Table 1.** Description of Territorial Context 1.

| **Territorial Context 1 (TC$_{\#1}$): Granada. Andalucía. Southern Spain: Warm and Temperate Climate** |
|---|
| The winter months are much rainier than the summer months. According to the Köppen classification the climate of this territory is classified as Mediterranean typical of warm summer (Csa). The average annual temperature is 15.5 °C, with average annual rainfall of 450 mm. The driest month is July, with 5 mm. Most of the precipitation in this territory falls in December, approximately 63 mm. The hottest month of the year is July with an average of 25.5 °C and January is the coldest month with average temperatures of 7.0 °C. The difference in precipitation between the driest month and the rainiest month is 58 mm. Throughout the year, temperatures vary around 18.5 °C. Average environmental humidity level. |

**Table 2.** Description of Territorial Context 2.

| **Territorial Context 2 (TC$_{\#2}$): Santiago de Compostela. Galicia. Northern Spain: Warm and Temperate Climate** |
|---|
| The vast majority of rain in this city falls in winter and is relatively scarce in summer. The climate classification of Köppen for this place is Mediterranean oceanic mild summer (Csb). The average temperature is 13.6 °C, the average annual rainfall is 1325 mm and the driest month is July, with 36 mm. In December, precipitation peaks, with an average of 191 mm/m$^3$. The hottest month of the year is August with an average temperature of 20.3 °C. January is the coldest month of the year with an average temperature of 8.7 °C. The difference in precipitation between the driest month and the rainiest month is 155 mm and the variation in annual temperature is around 11.6 °C. The difference in precipitation between the driest month and the rainiest month is 155 mm and the variation in annual temperature is around 11.6 °C. Average environmental humidity level. |

**Table 3.** Description of Territorial Context 3.

| **Territorial Context 3 (TC$_{\#3}$): Braga. North Portugal: Warm and temperate climate.** |
|---|
| There is more rainfall in winter than in summer and this location is included in the category of Mediterranean oceanic mild summer climate (Csb) in the Köppen classification. The average annual temperature is 14.2 °C and the average annual rainfall is approximately 1252 mm. The driest month is July, with 16 mm of rain. With an average of 170 mm, the rainiest month is December and July is considered the warmest month of the year with an average temperature of 20.3 °C. The rainiest month is December and July is considered the warmest month of the year with an average temperature of 20.3 °C. January has the lowest average temperature of the year, with 8.4 °C, and an estimated difference of 154 mm of precipitation between the driest and wettest months. During the year, average temperatures vary by 11.9 °C. The level of environmental humidity is average. |

### 3.2. Social Representations, Climate Literacy and Water

In 2015, the OECD produced a new PISA report that defines scientific literacy as "the ability to engage with science issues, and with the ideas of science, as a thoughtful citizen".

This definition presupposes that the scientific knowledge a person can attain will make him or her more likely to participate in reasoned discourses on science and that he or she will have the appropriate skills to be able to recognize, offer and evaluate explanations for a wide variety of natural and technological phenomena. On the other hand, it will be able to judge scientific questions by describing and evaluating the knowledge of this type available, as well as being able to interpret data in a variety of scientific representations, in order to ultimately draw appropriate conclusions [40].

According to the American Association for the Advancement of Science, climate literacy is part of science literacy because "science, math and technology have a profound impact on our individual lives and our culture. They play a role in almost all human efforts and affect the way we relate to each other and to the world around us... Science literacy enables us to make sense of real-world phenomena, informing people and making decisions and serves as the basis for a lifetime of learning" [41].

This conception of science literacy could be extrapolated to climate literacy, which would not cease to be science literacy. In a similar vein, the US Government's Global Change Research Program suggests that climate literacy should focus on an individual's understanding of the influence of a person on climate and the influence of climate on him or her and society at large. Thus, a climate literate person must understand the essential principles of the Earth and the climate system, know how to scientifically assess correct information about climate, communicate about climate and climate change in meaningful ways, and be able to be informed and make responsible decisions about actions that may affect it.

The importance of this approach to literacy lies in its civic dimension: society needs citizens who understand the climate system and know how to apply that knowledge in their daily lives, as well as exercising their commitment as active members of their communities, as the CC will continue to be a significant element of public becoming. Understanding the essential principles of climate science will enable people, for example, to evaluate media reports and critically argue their content.

Thus, climate literacy in society at large would make people more aware of the fundamental relationship between climate and life and would better reveal, for example, the many ways in which climate plays a fundamental role in human health [42].

This is why, although there is no specific concept for water literacy, since water is an indissoluble part of the climate system and is intimately related to the climate crisis, it can be included within this concept.

Orienting us on the object of study, university students, depending on their level of climate literacy, could generate different social representations on the CC. A social representation is defined as a particular modality of own knowledge of the common culture, whose functions are interpretative and also pragmatic, insofar as they serve to orient the behaviours and the communication between the individuals with respect to the represented object, in this case, the CC. Representation is an organized corpus of socially constructed and shared knowledge, beliefs, and valuations through which people make intelligible objects—in a broad sense, such as the CC—that are relevant to social reality. Representations can form part of the identity of a group or a society and modulate daily interactions, releasing the powers of individual and collective creativity, or cohorting them to the extent that they become hegemonic and naturalized [43].

Beyond this conceptualization, Farr emphasizes that social representations appear when individuals debate socially controversial issues or when they respond to the echo of events selected as significant by the media agenda. In addition, he adds that social representations have a double function: to make the strange familiar and the invisible perceptible, since the unusual or unknown can be personally threatening when one does not have a category to classify them. According to Farr, social representations are cognitive systems with their own logic and language. They do not simply represent opinions about images or attitudes towards an object, but they constitute profane theories or forms of knowledge with their own logic for the discovery, interpretation and organization of reality. They also contain systems of values, ideas and practices with a dual function: first, to establish a logical—common sense—order that allows individuals to orient themselves in their material and social world in order to appropriate and master it; second, to enable communication between members of a community by providing a code for social exchange and for naming and classifying without ambiguity (apparent) the various aspects of their world and of their individual and group history [44].

To understand that a social representation is a particular mode of knowledge, whose purpose is the elaboration of behaviours and communication among individuals, allows to face in a different way the appropriation of objects that come from the field of science by society in general, or by specific social groups, as well as to project from a new perspective the educational or communicative actions that are of public interest related to those objects. This is the case of the CC and its relationship with the problems it generates or power in relation to the availability and use of water resources. From the point of view of the common culture, this relationship can be understood in different ways, and will therefore generate different behaviours depending on the society and the social group considered, assuming that the understanding of the CC-water relationship and its interpretative and pragmatic repercussions are also different.

Communication based on knowledge of social perceptions and on the evaluation of the cultural and, in this case, geophysical contexts in which they take place, can facilitate people's acceptance of incorporating a whole repertoire of knowledge and sustainable actions into their lifestyle. Here, lifestyle is understood as the complex set of values, objectives, norms and activities, which also include ethical, environmental, economic and social presuppositions that regulate and concretize daily life [45].

There is a certain degree of uncertainty, as well as gaps in the research on the relationships between the CC and water [46]. However, there is also a significant degree of international and national concern about these relationships and, as a result, a great deal of research is underway, forming an increasingly solid body of basic knowledge [47].

Psychological studies on water sustainability have defined cognition as a deliberate, planned and systematic process in which beliefs (information available about the problem that can be considered false or true), perceptions (expectations about the consequences of water situations), motivations (extrinsic such as caring for water to save money or intrinsic such as conserving water so that children can use it in the future), attitudes (provisions relating to the assessment of situations or behaviours in the face of water scarcity and shortage), knowledge (management of information related to the water situation) and intentions (decisions of inaction or action in the face of water scarcity and shortage) seek to predict water expenditure or saving according to the relationships existing between the different variables involved [48]. There are multiple factors that depend on the construction of social representation on an object such as water. In the case of its relationship with the CC, research has been carried out where a minority of respondents point out thoughts related to the possible scarcity of water when they hear about climate change, due to the fact that due to the geographical situation of the individual or due to the hydrometeorological phenomena experienced, the perception of the abundance of water changes [49].

On the other hand, there are studies that show that the level of environmental awareness of each person is directly related to the degree of environmental perception, which is directly reflected in the environmental behaviour of the individual. Along these lines, studies on the environmental perception of university students have revealed that issues such as recycling, the economy of water and energy or the reduction of consumption are constructs that form environmental perception [50].

Some research with high school students has shown that, in general, incorrect ideas about the location and availability of fresh water on the planet predominate. In spite of the fact that it is a subject that is repeatedly addressed during compulsory secondary education, students are not aware of the number of people who currently do not have access to drinking water or who have a downward perception of their own water consumption in their daily activities [51].

*3.3. Water, CC and Sustainable Development Goals in Agenda 2030*

Water is the basis of life and an essential means of human subsistence, and is therefore key to sustainable development. Good water management translates into the achievement of many of the 17 SDG and particularly of SDG 6: "Ensure the availability and sustainable management of water and sanitation for all". Despite this aspiration, water is becoming a high-level social and geopolitical problem. So much so that up to 40% of the world's population will live in areas with severe water stress by 2035 and the capacity of ecosystems to provide water supplies will diminish. Bearing in mind that by 2050 the world's population is expected to reach 9.7 billion and that environmental and climatic conditions will become increasingly uncertain, transboundary water agreements should become more robust in order to secure water supplies for all people.

However, the perception of water as a human right, as a public good and as an environmental good, is very often opposed to its perception as a commodity that needs a price to be used. Furthermore, for this objective to be satisfactorily fulfilled, it must be understood that it is linked to two other essential SDG for sustainable development; SDG 13, "Action for Climate", and SDG 14, "Underwater Life". Adaptation to CC is essential for the protection of ecosystems and therefore for the development of human populations. Extreme weather events, rising sea levels or rising temperatures endanger water resources and threaten marine life and the quality of freshwater and ocean water, implying an increasingly unbalanced water cycle that hinders the human right to water supply.

## 4. Hypotheses

The perception of the CC, and in this case, the relationship it can have with water, can be conceived in different ways depending on the situation of the individual. We think that a person will not have the same perception of the lack of water in places where the precipitations are usually very abundant or where the vegetal mass is not scarce, therefore, it is possible that the variable of the territory influences in the social representation of the conception of this relation. On the other hand, the knowledge or level of climate literacy that the student believes he or she has about this relationship may also be affected by this variable depending on the place where it is socialized, as it is possible that in certain areas some environmental issues are given more importance than others. For this reason, in this study, we explore the different variables that may shed light on these issues, in order to check whether the territorial context is a factor that significantly influences the representation around the water/climate change relationship. If this is not the case, the degree of literacy that the individual has acquired is influenced by the information obtained regardless of where he or she is. The study has three hypotheses:

- $H_{\#1}$. The denial of the CC is significantly associated with a representation that belittles the consequences of global warming and other extreme phenomena.
- $H_{\#2}$. Territorial contexts with high average rainfall levels and low average annual temperatures tend to minimize the social representation of water risks associated with the CC; on the other hand, territorial contexts with low average rainfall levels and high average annual temperatures will tend to maximize the social representation of water effects and risks associated with the CC.
- $H_{\#3}$. The social representation on water and its relationship with the CC around its causes, consequences and solutions is created due to a significant relationship with the socio-cultural context.

## 5. Materials and Methods

### 5.1. Sample

The sample chosen for this study is a group of 1709 university students (61.3% were women and 38.4% men, with an average age around 21 years). 38.9% belonged to the branch of natural sciences and technologies and 61.1% to the branch of social sciences and humanities, distributed territorially as shown in Table 4.

**Table 4.** Sample Description.

| Territorial Context | Sample | % |
|:---:|:---:|:---:|
| TC$_{\#1}$ | 505 | 29.5% |
| TC$_{\#2}$ | 644 | 37.7% |
| TC$_{\#3}$ | 560 | 32.8% |
| TOTAL | 1709 | 100% |

### 5.2. Instrument

The instrument used for the collection of information is an ad hoc questionnaire that is divided into two blocks of Likert questions. The first block consists of questions whose content is true or false statements about the CC, designed to assess the student's knowledge on this subject. These statements are grouped into four dimensions (Table 5): causes of the CC, consequences of the CC, biophysical processes related to the CC and measures to combat the CC.

**Table 5.** Dimensions of the questionnaire and item Nº.

| Dimensions | Item Nº |
|---|---|
| Biophysical processes related to CC | 1, 4, 6, 8, 15 |
| Consequences of CC | 2, 3, 9, 10, 12, 14 |
| Causes of CC | 7 |
| Solutions to fight against the CC | 5, 11, 13, |

The answers to these questions present four options: Totally True (TT), Probably True (PT), Probably False (PF) and Totally False (TF). However, in order to make the results of the percentages of correct answers more concrete, we have chosen to consider totally and probably true as true (T), adding the percentages of both answers, and totally and probably false as false (F), performing the same procedure.

On the other hand, the second part of the questionnaire consists of questions more related to opinions, evaluations and personal perceptions. Tables 6 and 7 identifies the questions selected for this study and subsequently the selection criteria:

**Table 6.** Questionnaire questions considered Independent Variables.

**Independent Variables**

- To what extent do you feel informed about climate change in general?
- To what extent do you feel informed about the causes of climate change?
- To what extent do you feel informed about solution to fight against the climate change?
- To what extent do you feel informed about the consequences of climate change?
- Rate the climate change training you have received in your degree course
- Assess your degree of pro-environmental attitude
- Have you participated in any specific training activity related to climate change?

**Table 7.** Questionnaire questions considered Dependent Variables.

**Dependent Variables**

$V_{\#1}$. The greenhouse effect is a natural phenomenon.
$V_{\#2}$. A warmer planet will expand the area of incidence of tropical diseases.
$V_{\#3}$. The increase in temperatures will contribute to the occurrence of extreme atmospheric phenomena (cyclones, hurricanes, floods, etc.).
$V_{\#4}$. The polar hole of the ozone causes the melting of the poles.
$V_{\#5}$. If we stop emitting greenhouse gases, we will not be affected by climate change.
$V_{\#6}$. Acid rain is one of the causes of climate change.
$V_{\#7}$. Increased meat consumption contributes to climate change.
$V_{\#8}$. The greenhouse effect puts life on Earth at risk.
$V_{\#9}$. The sea level is rising due to the expansion of water due to the rise in temperature.
$V_{\#10}$. Climate change will decrease rainfall in my country.
$V_{\#11}$. If we stop emitting greenhouse gases, we will be less vulnerable to climate change.
$V_{\#12}$. Climate change will exacerbate desertification problems in the Iberian Peninsula.
$V_{\#13}$. Climate change would be reduced if we planted more trees.
$V_{\#14}$. Many islands and coastal areas will be submerged by climate change.
$V_{\#15}$. The greenhouse effect is caused by human activity.

These questions have been selected according to different factors that may be related to the water crises, described in Section 2.1. In the first place, all those questions of the matrix questionnaire that have to do with the greenhouse effect or with the emission of greenhouse gases have been included. They are included in this analysis since they are closely related to the alteration of the water cycle if the emissions of these gases increase. Other items are explored if students consider the greenhouse effect as negative, ignoring that it is a natural process that contributes to creating the thermal conditions for life on Earth as we know it, although its alteration can cause negative effects [52]. On the other

hand, we include those questions that refer to the biophysical processes that the common culture tends to erroneously relate to the CC, such as the destruction of the ozone layer or acid rain. Many of the studies reviewed have shown an erroneous relationship between these anomalies linked to the atmosphere and the CC, and in this concrene study, it is interesting to know if they are also related to the water problem. Another of the statements is related to meat consumption, and attempts to provide information on the perception of the food production model; although this statement does not expressly address issues related to water, this element is intrinsically linked to the processes of meat production, with its corresponding contamination and intensive consumption if it is not carried out in a sustainable manner.

Another group of questions included refers to the consequences of an increase in temperature in order to assess the extent to which students are informed about diseases, extreme atmospheric phenomena or alterations in sea level due to an increase in temperature.

The questions referring to the country's rainfall or desertification are particularly interesting for assessing whether there are differences in the perception of the CC in areas where the rainfall records are so different, bearing in mind that people living in a dry or wet end of the Iberian Peninsula are surveyed.

Finally, it incorporates a question that alludes to reforestation as a possible solution to the CC; like the question on meat consumption, no reference is made to water, but it can provide valuable information given that the students surveyed live in territories where the forest mass is very different and even to find out if they are informed about the intimate relationship that exists between the quantity of existing trees and the rainfall and humidity of an area (rainfall, losses due to runoff, evaporation...).

All the questions selected are relevant to know how social representation is generated around the CC and its relationship with water, and, above all, to assess the degree of climate literacy of university students, taking into account, a priori, that they may be considered to be scientifically educated people, with critical capacity to assess the information they receive and to contrast it with other sources.

Similarly, these questions have a solid scientific basis, corroborated by recent studies [53–65] that show evidence of the close relationship between the CC and the water cycle in the different critical vectors mentioned in Section 2.1.

*5.3. Instrument Quality Criteria and Data Analysis*

With regard to the reliability of the questionnaire, according to the Cronbach alpha coefficient is 0.74, and the validity of content has been considered by the frequency of statements that can be found in the different information and dissemination media about the CC (textbooks, internet, television, radio, dissemination media, etc.).

The analysis of the data has consisted of determining which questions grouped by causes, consequences, solution to figth against the CC and biophysical processes linked to the CC are answered correctly, that is to say, the questions in the first part of the questionnaire; and relating them to the degree of information that the individual claims to have and to his pro-environmental attitudes, questions that will be explored in the second part of the questionnaire. On the other hand, the results obtained are contrasted according to the territorial context of the student in order to determine whether there are significant differences with respect to the territorial variable. To this end, contingency tables have first been drawn up with the Spearman correlation test to identify whether there is a relationship between the variables that have been included in each category. Afterwards, the descriptive frequency statistics were calculated to determine the percentages of correct responses and, on observing a normal distribution of frequencies, the parametric tests were carried out to obtain the contrast of means, for which the analysis of the variance or ANOVA of a factor was carried out to compare the results between the different territorial contexts using the independent variables, the Chi-Square test was also carried out to obtain the statistically significant differences ($p < 0.05$) and a contrast of multiple post hoc comparisons (Scheffé Test), to determine which means differ from others. These parametric tests have been performed with the statistical quantitative software SPSS V25.0 (Statistical Package for Social Sciences, Armonk, NY, USA).

## 6. Results

The results obtained are presented below. Firstly, the results that refer to questions of personal opinion for the group of respondents and, on the other hand, for each of the contexts individually are shown. Next, the results are presented by context of those questions that were used to determine the degree of knowledge about the relationship between CC and its relationship with water, and finally, to relate them to the degree of information and pro-environmental attitude of the student.

97.9% of respondents think that CC is occurring, of which 85% believe that it is "mainly due to human causes", so that 2.1% of respondents can be considered to be negativists. However, it is curious the result of the question on the perception of the degree of agreement in the scientific community on this aspect: practically half of the respondents, 50.2%, think that there is agreement, while 49.5% perceive that there is not. With respect to the responsibility they believe their country of residence has in CC generation, 67.8% of respondents in the three selected contexts think that this responsibility exists; however, the perception of personal responsibility is somewhat lower, with 51.4%.

When the question refers to the perception of the degree to which CC affects them personally and the country, the students surveyed believe that it will affect them personally with a rather high percentage, 82.3%, but they believe that the effect on the country where they reside will be even greater, with 92.6% (Figure 5).

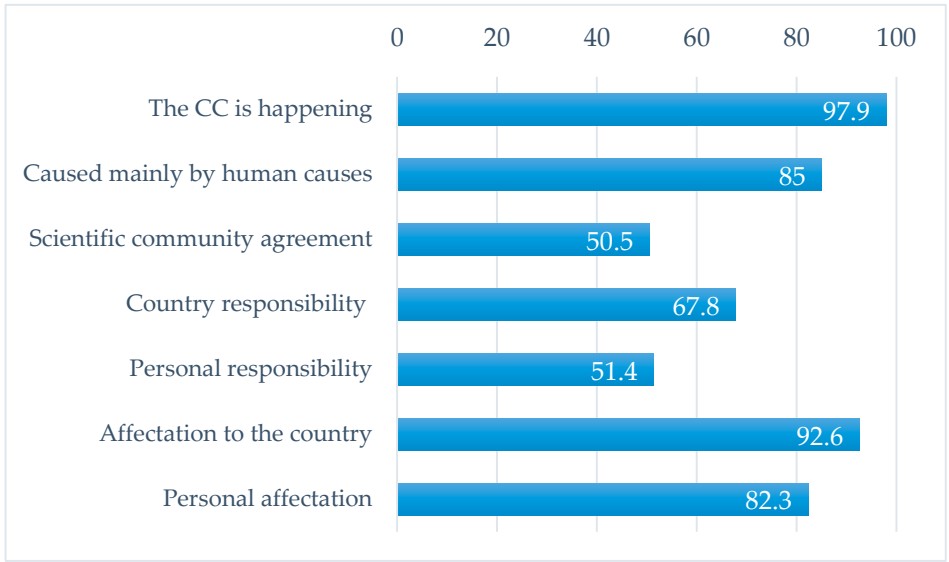

**Figure 5.** General opinion questions on CC.

Figure 6 shows the results for these same questions according to the territorial context. As can be observed, it is the students of the TC$_{\#3}$ who most believe that CC exists (99.3%), although with very little difference from the rest (TC$_{\#2}$ 98% and TC$_{\#1}$ 95.2%). They are also those who most attribute CC to human causes (TC$_{\#1}$ 80.4%, TC$_{\#2}$ 86.8% and TC$_{\#3}$ 87.1%) and those who with the highest percentage think that there is scientific consensus on this attribution (TC$_{\#1}$ 43.2%, TC$_{\#2}$ 49.3% and TC$_{\#3}$ 58.2%). However, the students in the TC$_{\#3}$ sample are also those who least believe that their country has responsibility in the climate crisis (TC$_{\#1}$ 74.4%, TC$_{\#2}$ 72.6% and TC$_{\#3}$ 55.9%) and those who attribute less responsibility to themselves in their causes (TC$_{\#1}$ 60%, TC$_{\#2}$ 58.2% and TC$_{\#3}$ 35.9%). Even so, the three contexts practically coincide in their totality, in that CC will affect their country, being the students of TC$_{\#2}$ who obtain the highest percentage (TC$_{\#1}$ 90.1%, TC$_{\#2}$ 95.2% and TC$_{\#3}$ 92%) but being those of TC$_{\#1}$ those who obtain the highest percentage when asked about the personal affectation before CC (TC$_{\#1}$ 84.5%, TC$_{\#2}$ 82.3% and TC$_{\#3}$ 74.4%).

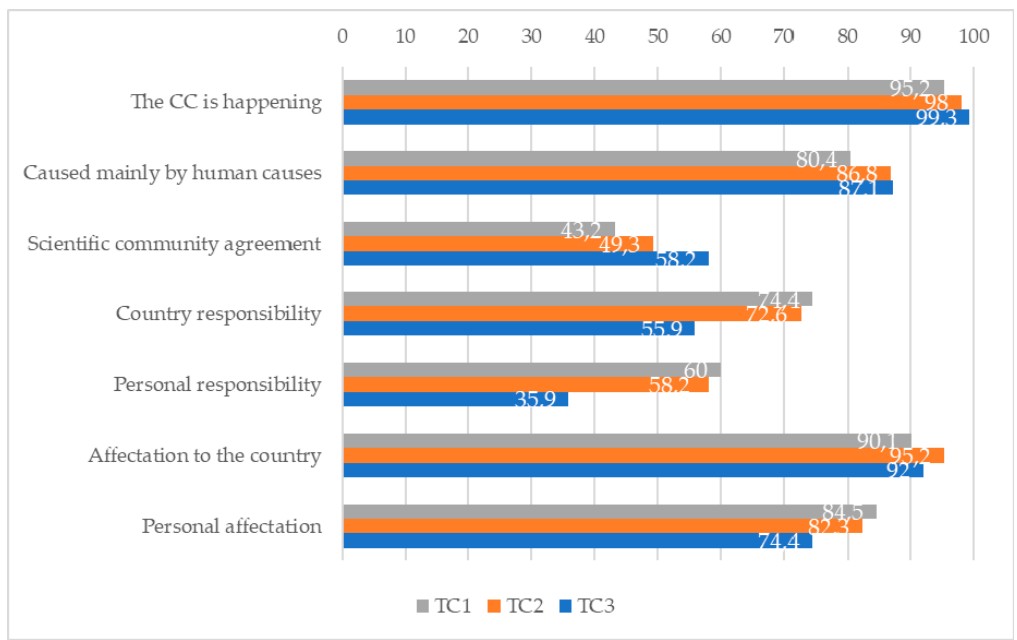

**Figure 6.** General opinion questions on CC in the different contexts analysed.

The correct response percentages, the sensation of information that the student thinks have with respect to different dimensions of CC and the degree of pro-environmental attitude of the same, are shown in the following Figures 7 and 8.

Table 8 shows the significant differences between the dimensions of CC in general. As can be seen, there are differences ($p < 0.05$) in all the dimensions studied except in the dimension of solutions. Specifically, in the biophysical processes dimension, TC$_{\#3}$ is the one that points out differences between the other two contexts studied. In the consequences dimension, the differences are found, however, in TC$_{\#2}$ with the other contexts. For the other dimensions no significant differences are found.

Below are the items, classified by dimensions, where statistically significant differences have been found between the contexts analysed (Table 9). As can be seen, in the dimension of biophysical processes related to CC there are significant differences in those questions related to the greenhouse effect and, specifically, question 1 has the greatest variability (F = 20.028), and TC$_{\#3}$ being the one that these differences are found. However, there are no differences in any context when dealing with questions related to the hole in the ozone layer or acid rain.

In the case of the consequences dimension of CC, we observe that it is the dimension where more variability of differences exists between the three territorial contexts. When reference is made to the desertification of the Iberian Peninsula due to CC we find differences in the three contexts and with a high variability (F = 47.564). On the other hand, there are also differences in this dimension between TC$_{\#1}$ and TC$_{\#2}$ and between TC$_{\#1}$ and TC$_{\#3}$ for questions 3, 9 and 14. For questions 2 and 10, the latter (question 10) with the greatest variability in this dimension (F = 75,847), the differences are found between TC$_{\#1}$ and TC$_{\#3}$ and between TC$_{\#2}$ and TC$_{\#3}$.

**Table 8.** Significant differences by items of dimensions of CC and TC.

| Dimension | Sig. between TC | | | | |
|---|---|---|---|---|---|
| | F | Sig. | TC$_{\#1}$-TC$_{\#2}$ | TC$_{\#1}$-TC$_{\#3}$ | TC$_{\#2}$-TC$_{\#3}$ |
| Biophysical processes related to CC | 8.572 | **0.000 \*\*** | 0.838 | **0.008 \*\*** | **0.000 \*\*** |
| Consequences of CC | 14.750 | **0.000 \*\*** | **0.000 \*\*** | 0.651 | **0.000 \*\*** |
| Causes of CC | 3.916 | **0.020 \*** | 0.990 | 0.053 | 0.053 |
| Solutions to fight against CC | 1.673 | 0.188 | 0.194 | 0.738 | 0.584 |

\*\* <0.01; \* < 0.05. Significant differences in bold.

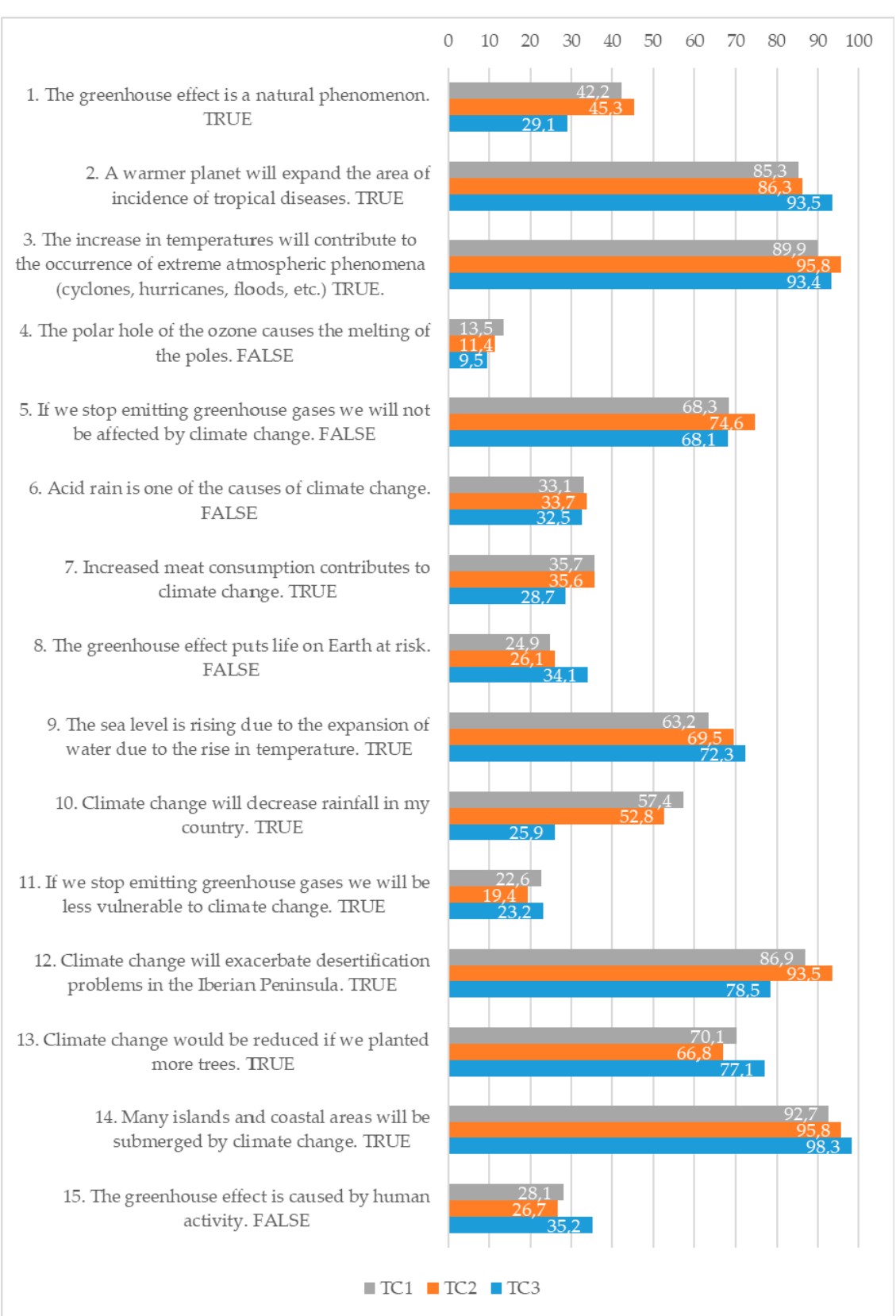

**Figure 7.** Conceptual questions about CC.

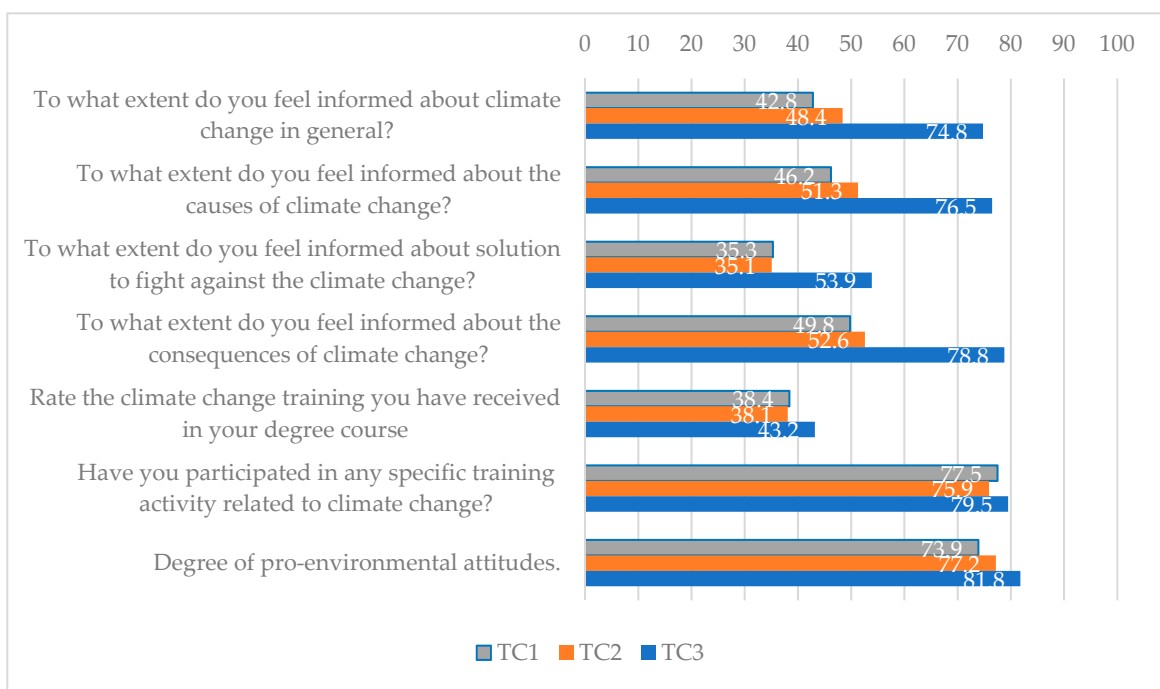

**Figure 8.** Degree of information about CC and pro-environmental attitudes.

**Table 9.** Significant differences by items of dimensions of CC and TC and correct response %.

| Dimension | Sig. between TC | | | | | Correct Response % | | |
|---|---|---|---|---|---|---|---|---|
| | F | Sig | TC#1-TC#2 | TC#1-TC#3 | TC#2-TC#3 | TC#1 | TC#2 | TC#3 |
| **Biophysical processes related to CC** | | | | | | | | |
| 1. The greenhouse effect is a natural phenomenon | 20.028 | **0.000 ** | 0.991 | **0.000 ** | **0.000 ** | 42.2 | 45.3 | 29.1 |
| 4. The polar hole of the ozone causes the melting of the poles | 1.363 | 0.256 | 0.852 | 0.274 | 0.530 | 13.5 | 11.4 | 9.5 |
| 6. Acid rain is one of the causes of climate change | 0.794 | 0.452 | 0.989 | 0.527 | 0.577 | 33.1 | 33.7 | 32.5 |
| 8. The greenhouse effect puts life on Earth at risk | 8.652 | **0.000 ** | 0.987 | **0.004 ** | **0.001 ** | 24.9 | 26.1 | 34.1 |
| 15. The greenhouse effect is caused by human activity | 8.652 | **0.000 ** | 0.446 | **0.031 *** | **0.000 ** | 28.1 | 26.7 | 35.2 |
| **Consequences of CC** | | | | | | | | |
| 2. A warmer planet will expand the area of incidence of tropical diseases | 13.151 | **0.000 ** | 0.434 | **0.000 ** | **0.001 ** | 85.3 | 86.3 | 93.5 |
| 3. The increase in temperatures will contribute to the occurrence of extreme atmospheric phenomena (cyclones, hurricanes, floods, etc.) | 5.429 | **0.004 ** | **0.007 ** | **0.039 *** | 0.873 | 89.9 | 95.8 | 93.4 |
| 9. The sea level is rising due to the expansion of water due to the rise in temperature | 6.462 | **0.002 ** | **0.020 *** | **0.003 ** | 0.751 | 63.2 | 96.5 | 72.3 |
| 10. Climate change will decrease rainfall in my country | 75.847 | **0.000 ** | 0.332 | **0.000 ** | **0.000 ** | 57.4 | 52.8 | 25.9 |
| 12. Climate change will exacerbate desertification problems in the Iberian Peninsula | 47.564 | **0.000 ** | **0.006 ** | **0.000 ** | **0.000 ** | 86.9 | 93.5 | 75.8 |
| 14. Many islands and coastal areas will be submerged by climate change | 12.769 | **0.000 ** | **0.000 ** | **0.000 ** | 0.977 | 92.7 | 95.8 | 98.3 |
| **Causes of CC** | | | | | | | | |
| 7. Increased meat consumption contributes to climate change | 3.916 | **0.020 ** | 0.990 | 0.053 | 0.053 | 35.7 | 35.6 | 28.7 |
| **Solutions to fight against CC** | | | | | | | | |
| 5. If we stop emitting greenhouse gases we will not be affected by climate change. | 2.720 | 0.066 | 0.446 | **0.031** | **0.000 ** | 68.3 | 74.6 | 68.1 |
| 11. If we stop emitting greenhouse gases we will be less vulnerable to climate change. | 4.934 | **0.007 ** | 0.119 | 0.685 | **0.010 ** | 22.6 | 19.4 | 23.2 |
| 13. Climate change would be reduced if we planted more trees. | 6.268 | **0.002 ** | 0.843 | **0.033 ** | **0.003 ** | 70.1 | 66.8 | 77.1 |

** < 0.01; * < 0.05. Significant differences in bold.

Finally, in the dimension of causes of CC there are no significant differences and in the dimension of solutions to fight against CC it is again the TC$_{\#3}$ that differs from the rest in questions 5 and 13. Therefore, in general, it is observed that the TC$_{\#3}$ is that it differs to a greater extent from the rest of the contexts with 8 of 15 questions with significant differences, being only 4 questions that have differences between the TC$_{\#1}$ and TC$_{\#2}$.

Next, the results obtained in the correct answers are related to the degree of information/pro-environmental attitudes that the student claims to have, that is, the existing relationship between the results of the two previous graphs and the significant differences found in said relationship.

*6.1. Degree of General Information on CC and Water*

As can be seen in Figure 8, the individuals who feel better informed about CC in general are those in the TC$_{\#3}$ with 74.8%, well above those belonging to TC$_{\#1}$ and TC$_{\#2}$, with TC$_{\#2}$ declaring itself less informed with 50.4%, followed by TC$_{\#1}$ with 56.9%; that is, half of those surveyed in TC$_{\#1}$ and TC$_{\#2}$ consider themselves informed and the other not.

As can be seen in Figure 7, despite the fact that the individuals in the TC$_{\#3}$ sample are the ones who believe they are most informed about CC in general, they answer eight of the 15 questions of the questionnaire erroneously, however, the respondents in TC$_{\#1}$ and TC$_{\#2}$ answer seven of 15 erroneously, but their sensation of information is lower, so that opinion is more in line with the results than that of the respondents in TC$_{\#3}$.

With regard to the questions referring to the greenhouse effect and greenhouse gases (1, 5, 8, 11 and 15), it is noted that in none of the terrestrial contexts is it clear exactly what it is, since a rather low percentage of respondents respond correctly to these questions. Specifically, only 45.3% of respondents in TC$_{\#2}$, 42.2% in TC$_{\#1}$ and 29.1% in TC$_{\#3}$ gave a good answer to the statement "The greenhouse effect is a natural phenomenon". On the other hand, they do not answer well to the question "The greenhouse effect puts at risk life on Earth" with 34.1% of correct answers in TC$_{\#3}$, 26.1% in TC$_{\#2}$ and 24.9% in TC$_{\#1}$. In addition, with respect to the item "The greenhouse effect is caused by human activity", only 35.2% of the individuals in the TC$_{\#3}$ are correct, followed by TC$_{\#1}$ with 28.1% and TC$_{\#2}$ with 26.7%.

On the other hand, it is worth mentioning that, although in the three territorial contexts there is a majority good response to consider true the statement "If we stop emitting greenhouse gases we will not be affected by climate change", with practically identical percentages in TC$_{\#3}$ and TC$_{\#1}$ (68.1% and 68.3% respectively) and with 74.6% in TC$_{\#2}$, the same does not happen with the statement "If we stop emitting greenhouse gases we will be less vulnerable to climate change", being TC$_{\#2}$ those who obtain the lowest percentage of success, 19.4%, followed by TC$_{\#1}$ with 22.6% and TC$_{\#3}$ with 23.2%; this discrepancy may indicate that although they intuit that the solution is not only to stop emitting these gases, they do not recognize the fact that to stop doing so supposes a great mitigation of CC, so, it is possible to think that they do not have very clear in what exactly consists the biophysical process of the greenhouse effect, and the benefits of a reduction of emissions of this type of gases.

As for those items that refer to problems that are usually related—in common culture—to climate change, but are neither a cause nor a consequence of the same, statements 4 and 6, "The hole in the ozone layer causes melting at the poles" and "Acid rain is one of the causes of climate change", respectively, it can be noted that in no territorial context is the majority correct, being the first of them the one with the lowest percentage of correct answers: the TC$_{\#3}$ sample is the one that registers the lowest percentage with a 9.5%, followed by TC$_{\#2}$, with 11.4%, and TC$_{\#1}$, with 13.5%. Statement 7 receives a more equal percentage of correct answers between the three contexts, although it is still a low percentage, TC$_{\#3}$ 32.5%, TC$_{\#1}$ 33.1% and TC$_{\#2}$ 33.7%.

However, it should be noted that all items that refer to consequences of CC (items 2, 3, 9, 10, 12 and 14) are correctly answered by a very high percentage of students in all three contexts, except item 10, "Climate change will decrease rainfall in my country", which is valued as a correct statement by 29.5% in TC$_{\#3}$, and however, item 12, "Climate change will exacerbate desertification problems in the Iberian Peninsula", is correctly assessed in this context by 78.5% of the sample, which means that there

is some confusion in these issues since they are intimately related, so that, once again, the sensation of information does not correspond with the reality of the answers provided.

On the other hand, item 7, "The increase in meat consumption contributes to climate change", is another item that registers a very low percentage of correct answers, with 28.7% of correct answers in $TC_{\#3}$, and practically the same percentage in $TC_{\#1}$ with 35.7% and $TC_{\#2}$ with 35.6%.

However, item 14, "Climate change would be reduced if we planted more trees", is correctly valued by a high percentage in the three contexts, 77.1% in $TC_{\#3}$, 70.1% in $TC_{\#1}$ and 66.8% in $TC_{\#2}$; but it should be noted that this issue can be included in the category of solutions to climate change, as well as statements referring to the reduction of greenhouse gas emissions, 5 and 11, the latter being erroneously valued by a majority, from which it can be deduced that there is also confusion in these terms.

With respect to the significant differences for this same variable, "To what extent do you feel informed about CC in general?" as can be seen in Table 8, of the 15 items we have analyzed, there are differences between eight of them in $TC_{\#1}$ and between seven of them in both $TC_{\#2}$ and $TC_{\#3}$.

Specifically, the items where the greatest statistically significant differences are recorded in $CT_{\#1}$ are item 7 ($p < 0.01$), $F_{(1, 505)} = 4.085$, item 2 ($p < 0.01$), $F_{(1, 505)} = 6.696$, item 13 ($p < 0.01$), $F_{(1, 505)} = 3.793$. The remaining items that show statistically significant differences with ($p < 0.05$) are items 1, 2, 11, 14 and 15, with $F_{(1, 505)} = 3.251$; $F_{(1, 505)} = 3.746$; $F_{(1, 505)} = 3.061$; $F_{(1, 505)} = 4{,}628$, and $F_{(1, 505)} = 3.255$, respectively.

With respect to the $CT_{\#2}$ sample, the questions with the greatest significant differences are found in items 2 ($p < 0.01$) $F_{(1,644)} = 5.660$, 5 ($p < 0.01$) $F_{(1, 644)} = 5.892$, 7 ($p < 0.01$) $F_{(1, 644)} = 5.482$, 12 ($p < 0.01$) $F_{(1, 644)} = 4.908$ and 13 ($p < 0.01$) $F_{(1, 644)} = 4.701$.

The items that follow with the greatest significant differences are 3 ($p < 0.05$) $F_{(1, 644)} = 2.984$ and 14 ($p < 0.05$) $F_{(1, 644)} = 2.797$.

Finally, for the $TC_{\#3}$ sample, the items where the greatest differences are found are item 1 ($p < 0.01$) $F_{(1, 560)} = 6.388$, item 2 ($p < 0.01$) $F_{(1, 560)} = 4.153$, item 3 ($p < 0.01$) $F_{(1, 560)} = 4.470$, item 7 ($p < 0.05$) $F_{(1, 560)} = 4.650$ and 12 ($p < 0.01$) $F_{(1, 560)} = 7.712$. They are followed by items 11 and 14 with ($p < 0.05$) $F_{(1, 560)} = 3.206$ and 2.913, respectively.

In general, it is observed that items 2, 7 and 12 (Table 10) accumulate the greatest statistically significant differences ($p < 0.01$) in the three territorial contexts analyzed, with item 12 being the only one that results in significant differences in the three contexts.

**Table 10.** To what extent do you feel informed about CC in general?

| ITEMS | $TC_{\#1}$ | $TC_{\#2}$ | $TC_{\#3}$ |
|---|---|---|---|
| | Sig./F | Sig./F | Sig./F |
| 1. The greenhouse effect is a natural phenomenon. | **0.022 */3.251** | 0.061/2.264 | **0.000 **/6.388** |
| 2. A warmer planet will expand the area of incidence of tropical diseases. | **0.033 */3.746** | **0.000 **/5.660** | **0.006 **/4.153** |
| 3. The increase in temperatures will contribute to the occurrence of extreme atmospheric phenomena (cyclones, hurricanes, floods, etc.). | 0.596/0.630 | **0.019 */2.984** | **0.004 **/4.470** |
| 4. The polar hole of the ozone causes the melting of the poles. | 0.761/0.389 | 0.347/1.117 | 0.168/1.688 |
| 5. If we stop emitting greenhouse gases we will not be affected by climate change. | 0.059/2.503 | **0.000 **/5.892** | 0.130/1.892 |
| 6. Acid rain is one of the causes of climate change. | 0.373/1.042 | 0.170/1.608 | 0.634/0.571 |
| 7. Increased meat consumption contributes to climate change. | **0.007 **/4.085** | **0.000 **/5.482** | **0.003 **/4.650** |
| 8. The greenhouse effect puts life on Earth at risk. | 0.150/1.780 | 0.227/1.415 | 0.739/0.420 |
| 9. The sea level is rising due to the expansion of water due to the rise in temperature. | 0.198/1.562 | 0.993/0.064 | 0.888/0.212 |
| 10. Climate change will decrease rainfall in my country. | 0.096/2.129 | 0.803/0.408 | 0.654/0.541 |
| 11. If we stop emitting greenhouse gases we will be less vulnerable to climate change. | **0.028 */3.061** | 0.176/1.586 | **0.023 */3.206** |
| 12. Climate change will exacerbate desertification problems in the Iberian Peninsula. | **0.000 **/6.696** | **0.001 **/4.908** | **0.000 **/7.712** |
| 13. Climate change would be reduced if we planted more trees. | **0.010 **/3.793** | **0.001 **/4.701** | 0.184/1.619 |
| 14. Many islands and coastal areas will be submerged by climate change. | **0.011 */4.628** | **0.025 */2.797** | **0.034 */2.913** |
| 15. The greenhouse effect is caused by human activity. | **0.022 */3.255** | 0.054/2.341 | 0.084/2.225 |

** < 0.01; * < 0.05. Significant differences in bold.

*6.2. Degree of Information on the Causes of CC and Its Relationship with Water*

　　When asked about the degree of information they believe they have regarding the causes of CC, it is observed, once again, that it is the students in the TC$_{\#3}$ sample who believe they are best informed in this respect (76.5%); however, as shown in Figure 7, despite having a sensation of fairly high information, the responses are incorrect in more than half of the items in the questionnaire, specifically in eight items out of 15 (Figure 7). In fact, the only statement in this category that corresponds to the dimension "causes of the CC" is item 7, which received a majority of incorrect answers in the three territorial contexts, being the one that obtains the lowest rate of correct answers in the TC$_{\#3}$ sample (28.7%); in such a way that the students in this context overestimate their degree of information on the causes of CC with respect to their sensation of information in this dimension. With respect to the TC$_{\#2}$ sample, it stands out that practically half of the students, 51.3%, consider themselves well informed about the causes, although item 7 is correctly valued only by 35.6% of the respondents. The students in the TC$_{\#1}$ sample value their level of information on the causes of CC below that of the other samples, with 46.2% considering themselves well informed; however, if this perception is related to the percentage of correct answers obtained, it is practically the same as in TC$_{\#2}$, 35.7%.

　　When significant differences are analyzed (Table 11), they are TC$_{\#1}$ and TC$_{\#2}$ where they appear in seven items, by five items in TC$_{\#3}$.

**Table 11.** To what extent do you feel informed about the causes of CC?

| ÍTEMS | TC$_{\#1}$ | TC$_{\#2}$ | CT$_{\#3}$ |
|---|---|---|---|
| | Sig./F | Sig./F | Sig./F |
| 1. The greenhouse effect is a natural phenomenon. | **0.003 \*\*/4.770** | 0.460/906 | **0.002 \*\*/4.211** |
| 2. A warmer planet will expand the area of incidence of tropical diseases. | **0.001 \*\*/5.483** | 0.111/1.885 | 0.239/1.381 |
| 3. The increase in temperatures will contribute to the occurrence of extreme atmospheric phenomena (cyclones, hurricanes, floods, etc.). | 0.201/1.548 | **0.003 \*\*/3.982** | **0.001 \*\*/1.388** |
| 4. The polar hole of the ozone causes the melting of the poles. | 0.830/0.294 | **0.009 \*\*/3.394** | 0.237/1.688 |
| 5. If we stop emitting greenhouse gases we will not be affected by climate change. | 0.582/0.651 | **0.049 \*/2.400** | 0.062/2.252 |
| 6. Acid rain is one of the causes of climate change. | 0.781/0.361 | 0.970/0.133 | 0.850/0.341 |
| 7. Increased meat consumption contributes to climate change. | **0.027 \*/3.082** | **0.004 \*\*/3.938** | 0.056/2.318 |
| 8. The greenhouse effect puts life on Earth at risk. | 0.120/1.955 | 0.092/2.009 | 0.191/0.1535 |
| 9. The sea level is rising due to the expansion of water due to the rise in temperature. | 0.198/1.562 | 0.196/1.516 | 0.525/0.801 |
| 10. Climate change will decrease rainfall in my country. | 0.413/0.956 | 0.661/0.602 | 0.909/0.251 |
| 11. If we stop emitting greenhouse gases we will be less vulnerable to climate change. | **0.025 \*/3.147** | 0.301/1.220 | 0.932/0.212 |
| 12. Climate change will exacerbate desertification problems in the Iberian Peninsula. | **0.001 \*\*/5.975** | **0.001 \*\*/4.741** | **0.000 \*\*/7.360** |
| 13. Climate change would be reduced if we planted more trees. | 0.094/2.141 | **0.000/6.355** | **0.018 \*\*/3.004** |
| 14. Many islands and coastal areas will be submerged by climate change. | **0.002 \*\*/50174** | **0.025 \*/2.797** | **0.026 \*\*/2.792** |
| 15. The greenhouse effect is caused by human activity. | **0.009 \*\*/3.910** | 0.611/0.672 | 0.180/1.573 |

\*\* < 0.01; \* < 0.05. Significant differences in bold.

　　Specifically in the TC$_{\#1}$ sample, the greatest differences ($p < 0.01$) are recorded in item 1, $F_{(1, 505)} = 4.770$, item 2, $F_{(1, 505)} = 5.483$, item 12, $F_{(1, 505)} = 5.975$, item 14 $F_{(1, 505)} = 5.174$ and item 15 $F_{(1, 505)} = 3.910$. The remaining differences with ($p < 0.05$) are found in items 7 $F_{(1, 505)} = 3.082$ and 11 $F_{(1, 505)} = 3.147$.

　　In the TC$_{\#2}$ sample, the greatest differences ($p < 0.01$) appear in item 3 $F_{(1, 644)} = 3.982$, 4 $F_{(1, 644)} = 3.394$, 7 $F_{(1, 644)} = 3.938$, 12 $F_{(1, 644)} = 4.741$ and 13 $F_{(1, 644)} = 6.355$. The only item with a difference ($p < 0.5$) is 5 $F_{(1, 644)} = 2.400$,

　　In the TC$_{\#3}$ sample, the greatest differences ($p < 0.01$) appear in items 3 $F_{(1, 560)} = 1.388$ and 12 $F_{(1, 560)} = 7.360$. The remaining significant differences ($p < 0.05$) appear in item 1 $F_{(1, 560)} = 4.211$, 13 $F_{(1, 560)} = 3.004$ and 14 $F_{(1, 560)} = 2.792$. In this block, it can be observed that the items with the greatest significant differences are 12 and 13, being 12, once again, where the greatest differences are reproduced in the three contexts.

### 6.3. Degree of Information about Solutions to Fight Against CC and Their Relationship with Water

As for the feeling of being informed about the solutions to fight against CC, it can be observed that, once again, it is the students of TC#3 who perceive themselves as best informed (Figure 8); but, in this case, of the three items included in this dimension (items 5, 11 and 13), the answer is mostly correct in item 5 (68.1%) and intem 13 (77.1%); item 11 is answered correctly only by 23.2% of the sample (Figure 7). The TC#1 and TC#2 samples register similar percentages in their self-perception of the information available on the measures to combat CC, with 35.3% and 35.1%, respectively; as in the TC#3 sample, only item 11 is correctly valued by a very low percentage of students: 19.4% in TC#2 and 22.6% in TC#1.

With respect to the significant differences between the responses to these items in the three contexts, it is noted that this block is the one that registers the least significant differences between the responses. Both in TC#1 and in TC#2 there are only significant differences in two items, being in TC#1 in items 1 and 2 ($p < 0.05$) and $F_{(1, 505)} = 3.115$ and $F_{(1, 505)} = 3.746$, respectively. In the case of TC#2, significant differences are found specifically in items 10 and 15, being for item 10 ($p < 0.05$) $F_{(1, 644)} = 2.730$ and for item 15 ($p < 0.01$) $F_{(1, 644)} = 3.841$. In the case of TC#3, the greatest differences are found in items 3.12 and 13 with ($p < 0.01$) and $F_{(1, 560)} = 4.761$, $F_{(1, 560)} = 4.104$ and $F_{(1, 560)} = 4.420$, respectively; and with ($p < 0.05$) significant differences are found in items 6 $F_{(1, 560)} = 2.445$ and 7 $F_{(1, 560)} = 2.446$ (Table 12).

**Table 12.** To what extent do you feel informed about solutions to fight against CC?

| ÍTEMS | TC#1 | TC#2 | TC#3 |
|---|---|---|---|
| | Sig./F | Sig./F | Sig./F |
| 1. The greenhouse effect is a natural phenomenon. | **0.026 \*/3.115** | 0.081/2.089 | 0.182/1.567 |
| 2. A warmer planet will expand the area of incidence of tropical diseases. | **0.033 \*\*/3.746** | 0.241/1.375 | 0.071/2.167 |
| 3. The increase in temperatures will contribute to the occurrence of extreme atmospheric phenomena (cyclones, hurricanes, floods, etc.). | 0.681/0.502 | 0.280/1.271 | **0.001 \*/4.761** |
| 4. The polar hole of the ozone causes the melting of the poles. | 0.443/0.896 | 0.394/1.025 | 0.233/1.399 |
| 5. If we stop emitting greenhouse gases we will not be affected by climate change. | 0.891/0.208 | 0.073/2.149 | 0.817/0.388 |
| 6. Acid rain is one of the causes of climate change. | 0.798/0.337 | 0.259/1.327 | **0.046 \*\*/2.445** |
| 7. Increased meat consumption contributes to climate change. | 0.368/1.056 | 0.218/1.443 | **0.046 \*\*/2.446** |
| 8. The greenhouse effect puts life on Earth at risk. | 0.063/1.955 | 0.208/1.474 | 0.429/0.960 |
| 9. The sea level is rising due to the expansion of water due to the rise in temperature. | 0.843/276 | 0.379/1.053 | 0.670/0.590 |
| 10. Climate change will decrease rainfall in my country. | 0.197/1.564 | **0.028 \*\*/2.730** | 0.473/0.884 |
| 11. If we stop emitting greenhouse gases we will be less vulnerable to climate change. | 0.486/0.815 | 0.875/0.305 | **0.130**/1.789 |
| 12. Climate change will exacerbate desertification problems in the Iberian Peninsula. | 0.383/1.020 | 0.256/1.335 | **0.003 \*\*/4.104** |
| 13. Climate change would be reduced if we planted more trees. | 0.328/1.152 | 0.116/1.858 | **0.002 \*\*/4.420** |
| 14. Many islands and coastal areas will be submerged by climate change. | 0.358/4.628 | 0.502/0.836 | 0.144/1.719 |
| 15. The greenhouse effect is caused by human activity. | 0.105/1.079 | **0.004 \*\*/3.841** | 0.276/1.283 |

\*\* < 0.01; \* < 0.05. Significant differences in bold.

### 6.4. Degree of Information on the Consequences of the CC and Its Relationship with Water

In this block, it is once again the TC#3 sample that gathers the students with the greatest self-perception of being well informed about the consequences of CC (Figure 8), with 78.8% expressing this way, followed by the TC#2 sample, with 52.6% and the TC#1 sample, with 49.8%. Following previous patterns, the TC#3 sample continues to be the one that responds mostly incorrectly to more respondents, specifically eight out of 15. The other samples do not differ much from this pattern, since they erroneously value seven out of 15 items, but the self-perception of their level of information regarding the consequences of CC is more in line with the number of well valued items. However, in all the items of this dimension (2, 3, 9, 10, 12 and 14), about the consequences of CC, the three contexts add up majority percentages of correct answers, except in item 10 in TC#3 (Figure 7).

In the case of statistically significant differences (Table 13), the context with the smallest differences between responses is the $TC_{\#1}$: they are found with ($p < 0.01$) in items 3, 12 and 13, with $F_{(1,\,505)} = 3.746$, $F_{(1,\,505)} = 4.987$ and $F_{(1,\,505)} = 3.793$, respectively.

**Table 13.** To what extent do you feel informed about the consequences of the CC?

| ÍTEMS | $TC_{\#1}$ | $TC_{\#2}$ | $TC_{\#3}$ |
|---|---|---|---|
| | Sig./F | Sig./F | Sig./F |
| 1. The greenhouse effect is a natural phenomenon. | 0.057/2.525 | **0.002 **/4.292 | **0.002 **/4.414 |
| 2. A warmer planet will expand the area of incidence of tropical diseases. | **0.033 */3.746 | 0.057/2.307 | **0.018 */3.000 |
| 3. The increase in temperatures will contribute to the occurrence of extreme atmospheric phenomena (cyclones, hurricanes, floods, etc.). | 0.005/4.363 | 0.200/1.501 | **0.000 **/6.743 |
| 4. The polar hole of the ozone causes the melting of the poles. | 0.635/0.570 | 0.118/1.846 | **0.014 */3.156 |
| 5. If we stop emitting greenhouse gases we will not be affected by climate change. | 0.506/0.780 | **0.020 **/2.954 | 0.117/1.855 |
| 6. Acid rain is one of the causes of climate change. | 0.469/0.847 | 0.065/2.223 | 0.762/0.464 |
| 7. Increased meat consumption contributes to climate change. | **0.030 **/3.002 | **0.007 */3.545 | 0.296/1.231 |
| 8. The greenhouse effect puts life on Earth at risk. | 0.44/0.893 | 0.115/1.864 | 0.190/1.538 |
| 9. The sea level is rising due to the expansion of water due to the rise in temperature. | 0.528/0.742 | 0.623/0.656 | 0.790/0.425 |
| 10. Climate change will decrease rainfall in my country. | 0.200/1.552 | 0.356/1.098 | 0.677/0.580 |
| 11. If we stop emitting greenhouse gases we will be less vulnerable to climate change. | 0.473/0.838 | 0.661/0.602 | 0.397/1.019 |
| 12. Climate change will exacerbate desertification problems in the Iberian Peninsula. | **0.002 **/4.987 | **0.000 **/5.570 | **0.000 **/6.179 |
| 13. Climate change would be reduced if we planted more trees. | **0.010 **/3.793 | **0.001 **/4.563 | 0.072/2.161 |
| 14. Many islands and coastal areas will be submerged by climate change. | 0.084/2.227 | **0.034 */2.623 | **0.025 */2.820 |
| 15. The greenhouse effect is caused by human activity. | 0.137/1.854 | **0.029 */2.707 | **0.031 */2.683 |

\*\* <0.01; \* < 0.05. Significant differences in bold.

Differences with ($p < 0.05$) appear in item 2, $F_{(1,\,505)} = 3.746$, and in item 7 $F_{(1,\,505)} = 3.002$. In $TC_{\#2}$, the greatest differences ($p < 0.01$) are recorded in items 1 $F_{(1,\,644)} = 4.292$, 7 $F_{(1,\,644)} = 3.545$, 12 $F_{(1,\,644)} = 5.570$ and 13 $F_{(1,\,644)} = 4.563$. With ($p < 0.05$) 5 $F_{(1,\,644)} = 2.954$, 14 $F_{(1,\,644)} = 2.623$ and 15 $F_{(1,\,644)} = 2.707$ appear in items 5 $F_{(1,\,644)} = 2.623$ and 15 $F_{(1,\,644)} = 2.707$. In the case of $TC_{\#3}$, the greatest differences ($p < 0.01$) are in items 1 $F_{(1,\,560)} = 4.414$, 3 $F_{(1,\,650)} = 6.743$ and 12 $F_{(1,\,560)} = 6.179$. The differences ($p < 0.05$) appear in items 2 $F_{(1,\,560)} = 3.0$, 4 $F_{(1\,560)} = 3.156$, 14 $F_{(1,\,650)} = 2.820$ and 15 $F_{(1,\,650)} = 2.638$. It can also be observed that item 12 repeats in the three contexts the greatest significant differences.

*6.5. Assessment of the Degree of Information Received about CC in Your Degree*

For the question *"Values the training received on CC"*, the samples of the three contexts considered coincide mostly in not having received sufficient training on CC in their degree; in fact, they only consider that this has been the case for 28.2% in $TC_{\#1}$, 38.1% in $TC_{\#2}$ and 43.2% in $TC_{\#3}$ (Figure 5).

The greatest significant differences (Table 14) ($p < 0.01$) for this case, in $TC_{\#1}$, are found in items 1 $F_{(1,\,505)} = 10.259$, 6 $F_{(1,\,505)} = 3.620$ and 15 $F_{(1,505)} = 4.281$. With ($p < 0.05$) significant differences are recorded in items 2 $F_{(1,\,505)} = 3.746$, 5 $F_{(1,\,505)} = 2.598$, 9 $F_{(1,\,505)} = 2.709$ and 12 $F_{(1,505)} = 2.506$. In the case of $TC_{\#2}$, the greatest differences appear in items 1 $F_{(1,\,644)} = 4.111$, 5 $F_{(1,\,644)} = 3.503$, 7 $F_{(1,\,644)} = 3.808$, 10 $F_{(1,\,644)} = 3.147$ and 13 $F_{(1,\,644)} = 6.075$. Other differences ($p < 0.05$) are found in items 4 $F_{(1,\,644)} = 2.422$ and 14 $F_{(1,\,644)} = 2.797$. For this question, the sample of $TC_{\#3}$ registers a smaller number of significant differences between the answers, being item 15 the most outstanding with ($p < 0.01$) $F_{(1,\,560)} = 3.030$, and with ($p < 0.05$) item 1 $F_{(1,\,560)} = 2.675$ and item 8 $F_{(1,\,560)} = 2.809$.

In this case, item 1 is the only one that shows significant differences in the three contexts.

**Table 14.** Assesses the degree of information received on CC in your degree.

| ÍTEMS | TC#1 | TC#2 | TC#3 |
|---|---|---|---|
| | Sig./F | Sig./F | Sig./F |
| 1. The greenhouse effect is a natural phenomenon. | **0.000 **/10.259** | **0.001 **/4.111** | **0.021 */2.675** |
| 2. A warmer planet will expand the area of incidence of tropical diseases. | **0.033 */3.746** | 0.218/1.411 | 0.861/0.383 |
| 3. The increase in temperatures will contribute to the occurrence of extreme atmospheric phenomena (cyclones, hurricanes, floods, etc.). | 0.737/0.498 | 0.162/1.585 | 0.286/1.246 |
| 4. The polar hole of the ozone causes the melting of the poles. | 0.753/0.477 | **0.034 */2.422** | 0.676/0.632 |
| 5. If we stop emitting greenhouse gases we will not be affected by climate change. | **0.036 */2.598** | **0.004 **/3.503** | 0.682/0.682 |
| 6. Acid rain is one of the causes of climate change. | **0.006 **/3.625** | 0.172/1.552 | 0.444/0.957 |
| 7. Increased meat consumption contributes to climate change. | 0.067/2.211 | **0.00 */8.308** | 0.496/0.877 |
| 8. The greenhouse effect puts life on Earth at risk. | **0.012*/3.237** | 0.969/0.182 | **0.016 */2.809** |
| 9. The sea level is rising due to the expansion of water due to the rise in temperature. | **0.030 */2.709** | 0.428/0.982 | 0.433/0.973 |
| 10. Climate change will decrease rainfall in my country. | 0.582/0.715 | **0.008 */3.147** | 0.182/1.520 |
| 11. If we stop emitting greenhouse gases we will be less vulnerable to climate change. | 0.596/0.694 | 0.549/0.801 | 0.413/1.006 |
| 12. Climate change will exacerbate desertification problems in the Iberian Peninsula. | **0.041 */2.506** | 0.161/1.588 | 0.219/1.411 |
| 13. Climate change would be reduced if we planted more trees. | 0.690/0.562 | **0.000 */6.075** | 0.353/1.111 |
| 14. Many islands and coastal areas will be submerged by climate change. | 0.153/1.679 | **0.025 */2.797** | 0.526/0.834 |
| 15. The greenhouse effect is caused by human activity. | **0.002 **/4.281** | 0.038/2.370 | **0.010 **/3.030** |

** < 0.01; * < 0.05. Significant differences in bold.

## 6.6. Assessment of the Degree of Pro-Environmental Attitude

In this case, the sample with the greatest pro-environmental attitude is TC#3, with 81.8%, followed by TC#2, with 77.2%, and TC#1 with 73.9% (Figure 8).

The greatest significant differences ($p < 0.01$) in this question (Table 15) are evident for TC#1 in items $F_{(1,505)} = 4.819$, 10 $F_{(1,505)} = 3.740$, 12 $F_{(1,505)} = 4.182$ and 14 $F_{(1,505)} = 4.329$. The differences ($p < 0.05$) appear in items 2 $F_{(1,505)} = 3.746$, 11 $F_{(1,505)} = 3.050$ and 13 $F_{(1,505)} = 2.510$.

**Table 15.** Assess your degree of pro-environmental attitude.

| ÍTEMS | TC#1 | TC#2 | TC#3 |
|---|---|---|---|
| | Sig./F | Sig./F | Sig./F |
| 1. The greenhouse effect is a natural phenomenon. | 0.085/2.056 | 0.444/0.957 | **0.029 */2.729** |
| 2. A warmer planet will expand the area of incidence of tropical diseases. | **0.033/3.746** | **0.017 */2.768** | **0.006/3.607** |
| 3. The increase in temperatures will contribute to the occurrence of extreme atmospheric phenomena (cyclones, hurricanes, floods, etc.). | 0.553/0.759 | **0.000 **/5.369** | **0.020 */2.948** |
| 4. The polar hole of the ozone causes the melting of the poles. | 0.120/1.838 | 0.109/1.810 | 0.646/0.624 |
| 5. If we stop emitting greenhouse gases we will not be affected by climate change. | 0.367/1.077 | 0.312/1.192 | 0.742/0.491 |
| 6. Acid rain is one of the causes of climate change. | 0.251/1.349 | 0.720/0.573 | 0.237/1.387 |
| 7. Increased meat consumption contributes to climate change. | **0.001 **/4.819** | **0.11 */3.019** | 0.081/2.090 |
| 8. The greenhouse effect puts life on Earth at risk. | 0.862/0.324 | 0.796/0.474 | 0.094/1.991 |
| 9. The sea level is rising due to the expansion of water due to the rise in temperature. | 0.715/0.528 | 0.235/1.365 | 0.087/2.042 |
| 10. Climate change will decrease rainfall in my country. | **0.006 **/3.704** | 0.116/1.774 | 0.769/0.454 |
| 11. If we stop emitting greenhouse gases we will be less vulnerable to climate change. | **0.017 */3.050** | 0.090/1.911 | 0.736/0.500 |
| 12. Climate change will exacerbate desertification problems in the Iberian Peninsula. | **0.002 **/4.182** | **0.000 **/6.098** | **0.000 **/7.638** |
| 13. Climate change would be reduced if we planted more trees. | **0.041 */2.510** | **0.000 **/4.493** | **0.008 **/3.506** |
| 14. Many islands and coastal areas will be submerged by climate change. | **0.002/4.329** | **0.000/4.989** | **0.023/2.857** |
| 15. The greenhouse effect is caused by human activity. | 0.231/1.406 | 0.075/2.014 | **0.006/3.615** |

** < 0.01; * < 0.05. Significant differences in bold.

In the TC$_{\#2}$ sample, the greatest differences ($p <0.01$) are recorded in items 3 F$_{(1, 644)}$ = 5.369, 7 F$_{(1, 644)}$ = 3.019, 12 F$_{(1, 644)}$ = 6.098, 13 F$_{(1, 644)}$ = 4.493 and 14 F$_{(1, 640)}$ = 4.989. The differences ($p < 0.05$) appear in items 2 F$_{(1, 644)}$ = 2.768 and 7 F$_{(1, 644)}$ = 3.019.

In the TC$_{\#3}$ sample, the greatest significant differences ($p < 0.01$) are recorded in items 2 F$_{(1, 560)}$ = 3.607, 12 F$_{(1, 560)}$ = 7.638, 13 F$_{(1, 560)}$ = 3.506 and 15 F$_{(1, 560)}$ = 3.615. Differences ($p < 0.05$) appear in items 1 F$_{(1, 560)}$ = 2.729, 3 F$_{(1, 560)}$ = 2.948 and 14 F$_{(1, 560)}$ = 3.506. Again, it is item 12 that shows statistically significant differences in the three contexts.

## 6.7. Participation in CC-Related Activities

Finally, as can be seen in Figure 8, the vast majority of those surveyed have participated in some specific training activity related to CC, with TC$_{\#3}$ students recording the highest percentage, with 79.5%, followed by TC$_{\#1}$, with 77.5%, and TC2, with 75.9%. In the case of the significant differences (Table 16), in the TC$_{\#1}$ sample there are only two items with significant differences, 7, with the greatest difference, ($p < 0.01$) F$_{(1, 505)}$ = 13.977 and item 2 ($p < 0.05$) F$_{(1, 505)}$ = 3.746.

**Table 16.** Have you participated in any specific training activity related to the CC?

| ÍTEMS | TC$_{\#1}$ | TC$_{\#2}$ | TC$_{\#3}$ |
|---|---|---|---|
| | Sig./F | Sig./F | Sig./F |
| 1. The greenhouse effect is a natural phenomenon. | 0.222/1.497 | 0.199/1.617 | **0.041** */3.224 |
| 2. A warmer planet will expand the area of incidence of tropical diseases. | **0.033** */3.746 | 0.569/0.564 | **0.001** **/7.332 |
| 3. The increase in temperatures will contribute to the occurrence of extreme atmospheric phenomena (cyclones, hurricanes, floods, etc.). | 0.142/2.160 | **0.021** */3.888 | **0.027** */3.635 |
| 4. The polar hole of the ozone causes the melting of the poles. | 0.451/0.568 | 0.771/0.261 | 0.841/0.174 |
| 5. If we stop emitting greenhouse gases we will not be affected by climate change. | 0.585/0.298 | 0.536/0.625 | 0.254/1.375 |
| 6. Acid rain is one of the causes of climate change. | 0.239/1.390 | 0.128/2.064 | 0.572/0.560 |
| 7. Increased meat consumption contributes to climate change. | **0.000** **/13.977 | **0.04** **/5.621 | **0.015** */4.207 |
| 8. The greenhouse effect puts life on Earth at risk. | 0.419/0.654 | 0.476/0.744 | 0.957/0.044 |
| 9. The sea level is rising due to the expansion of water due to the rise in temperature. | 0.135/2.244 | 0.296/1.220 | **0.001** **/6.879 |
| 10. Climate change will decrease rainfall in my country. | 0.052/3.808 | 0.411/0.891 | 0.461/0.775 |
| 11. If we stop emitting greenhouse gases we will be less vulnerable to climate change. | 0.688/0.161 | 0.434/0.837 | 0.158/1.851 |
| 12. Climate change will exacerbate desertification problems in the Iberian Peninsula. | 0.094/2.822 | **0.033** */3.426 | **0.014** */4.311 |
| 12. Climate change will exacerbate desertification problems in the Iberian Peninsula. | 0.095/2.804 | 0.123/2.102 | 0.221/1.512 |
| 14. Many islands and coastal areas will be submerged by climate change. | 0.141/2.172 | 0.196/1.635 | 0.331/1.109 |
| 15. The greenhouse effect is caused by human activity. | 0.334/0.936 | 0.138/1.984 | **0.030** */3.531 |

** < 0.001; * < 0.05. Significant differences in bold.

In the TC$_{\#2}$ sample there are differences ($p < 0.01$) also in item 7 F$_{(1, 644)}$ = 5.62 and, with differences ($p < 0.05$), items 3 F$_{(1, 644)}$ = 3.888 and 12 F$_{(1, 644)}$ = 3.426.

However, more differences are evident in the TC$_{\#3}$ sample than in the others. The largest ($p < 0.01$) are found in item 2 F$_{(1, 560)}$ = 7.332 and 9 F$_{(1, 560)}$ = 6.879. Significant differences ($p < 0.05$) appear in item 1 F$_{(1, 560)}$ = 3.224, 3 F$_{(1, 560)}$ = 3.635, 7 F$_{(1, 560)}$ = 4.207, 12 F$_{(1, 560)}$ = 4.311 and 15 F$_{(1, 560)}$ = 3.531. Item 7 shows significant differences in any context.

## 7. Discussion

Based on the three hypotheses presented in this work, the results obtained are indicated:

H$_{\#1}$. Denying the existence of CC as a scientific phenomenon significantly affects the downward representation of the consequences of Global Warming and other extreme phenomena.

As can be seen, practically all university students surveyed in the three territorial contexts agree that CC exists, which can be compared with the results obtained in another study [66], where Spanish people of different ages, educational levels, habitat or ideology coincide at similar rates with this study

in that the CC is real and that it is caused mainly by human causes. This gives an idea that denialism is a downward trend, and in this specific case, it can be concluded that the $H_{\#1}$ hypothesis is not accepted since denying the existence of CC as a scientific phenomenon does not generate an underestimation of the consequences of global warming and other extreme phenomena, and therefore should not affect the social representations that are being generated in relation to the social acceptance of CC, since a minimum part of the sample studied denies the existence of CC.

On the other hand, the aforementioned study also coincides in percentage with those who think in our samples that there is no agreement among the scientific community, which is a very interesting data to know how the social representations of CC are being conformed, when it is well known that in this matter the scientific consensus is practically unanimous. That practically half of the students perceive divisions in the scientific community, but at the same time accept that climate change exists, implies that their belief is not so much based on scientific arguments or scientifically legitimized but on others that may be more volatile from an ideological, polygamous or cultural point of view. In other words, whoever believes almost as a matter of faith may cease to believe if other arguments question that belief.

In addition, the perception of the students surveyed regarding individual responsibility for CC is much lower than that which they generically attribute to the country in which they live. This discrepancy can be interpreted in terms of environmental hyperopia, that is, the problem does not feel psychologically close either in time or space, a pattern that seems to be replicated in different studies [67,68]: people tend to externalize their responsibility for the causes of CC—and their vulnerability to consequences—deriving that responsibility to the activity of companies, the role of governments or emissions from other countries. However, in this case, it can be observed that the students of the $TC_{\#3}$ are those who by far value their individual responsibility as the lowest, as well as the responsibility of their country of residence.

$H_{\#2}$. Territorial contexts with high average rainfall levels and low average annual temperatures exert a minimizing influence on the social representation of the effects and risk perception of CC, so that, on the contrary, territorial contexts with low average rainfall levels and high average annual temperatures will exert a maximizing influence on the social representation of the effects and risk perception of CC.

Statements referring to the effects of CC and the perception of the risk generated are considered to be those included in the consequences dimension, and specifically, those referring to the increase in temperature (items 2 and 3) and to rainfall and desertification in the Iberian Peninsula (items 10 and 12). As can be observed, the sample from $TC_{\#3}$ is the one that is most self-reported in this respect; however, it is the sample that obtains the lowest values of success in its responses. In fact, the students surveyed in this context are the only ones who mostly value item 10 incorrectly: "Climate change will decrease rainfall in my country". Therefore, the $H_{\#2}$ hypothesis can be rejected in this case, since the social representation of CC in relation to the perception of risk posed by its effects does not seem to depend on the specific territorial contexts considered here. In other words, the fact that a student belongs to a territory where rainfall is more abundant and the average annual temperature is lower than in another does not minimise his perception of the effects of the CC, since, although $TC_{\#3}$ answers item 10 incorrectly, all the others receive accurate assessments in the three contexts.

$H_{\#3}$. Different political-cultural contexts between two territories generate different pro-environmental attitudes and sensations of information and therefore different social representations on the causes, consequences and solutions of climate change and its relationship with water.

When analysing the results in each of the categories, it should be noted that such a similar response pattern has been generated that these categories could be classified into three different groups in relation to the self-perception of the level of information on CC:

- TC$_{\#3}$ has greater self-perception of its level of information, followed by Territorial TC$_{\#2}$ and TC$_{\#1}$ in the following variables:

  V$_{\#1}$: To what extent do you feel informed about climate change in general?
  V$_{\#2}$: To what extent do you feel informed about the causes of climate change?
  V$_{\#4}$: To what extent do you feel informed about the consequences of climate change?

- TC$_{\#3}$ has greater self-perception of its level of information than TC$_{\#1}$ and TC$_{\#2}$ and, which are equal in this self-perception, in the following variables:

  V$_{\#3}$: To what extent do you feel informed about measures to combat climate change?
  V$_{\#5}$: Rate the training received on climate change
  V$_{\#6}$: Assess your degree of pro-environmental attitude

- The student samples from the three territorial contexts considered equal their percentages in the following variable:

  V$_{\#7}$: Have you participated in any specific training activity related to climate change?

It is found that the students surveyed at TC$_{\#3}$ consider that they are more informed about CC in general, and specifically, about its causes and consequences, followed, in percentage order, of samples TC$_{\#2}$ and TC$_{\#1}$. On the other hand, the sample of TC$_{\#3}$ also perceives itself more informed about solution to fight against CC, with more training received in the degree and with greater pro-environmental attitudes than TC$_{\#1}$ and TC$_{\#2}$. However, the latter coincide in percentage.

Bearing in mind that TC$_{\#3}$ provides the contrast sample, with a territory with similar rainfall and temperature characteristics to those of TC$_{\#2}$, it could be said that the H$_{\#3}$ hypothesis is not accepted: given that, in relation to the sensation of information, different political-cultural contexts between two territories generate different self-perceptions of the level of information but not different social representations on the causes, consequences and solutions to the climate crisis and its relationship with water. In this sense, it should be noted that the levels of success in the three territorial contexts are very similar and concentrate on the same items.

In the case of items that refer to related biophysical processes, such as those related to the greenhouse effect and greenhouse gases, the existing confusion about them is evident and, in addition, the low success rate in the valuation of these statements follows a pattern already contacted in other studies [69–73]. There is also repeated confusion in the three contexts with those items that allude to issues related to the ozone layer or acid rain, a pattern that, again, is repeated in other research with samples of students of all ages [74].

On the other hand, the behaviour of the TC$_{\#3}$ differs from that of the TC$_{\#2}$ and TC$_{\#1}$ (which practically coincide in percentages) with respect to the appreciation of the training received on CC in their degree; even so, the self-perception they have with respect to the information they have received on CC in their degree is very low in the three contexts, data that also coincide, for example, in a study carried out in Gran Canarias [75]; it would be interesting to take this data into account for comparisons with other territories where this issue is a priority in the degrees and to assess how this variable influences the social representation of CC.

It should be borne in mind that when students are asked about meat consumption and their contribution to climate change, in all three contexts there is a lack of information on the subject, so that, like the items related to the greenhouse effect or greenhouse gases, the information deficit can help generate a social representation of CC that is out of tune with the reality and severity of the problem.

In the case of the statements on solutions to fight against the CC, such as those that speak of reducing emissions or planting more trees, in the three territorial contexts studied there is agreement on their veracity, as in the aforementioned study [67].

It is necessary to emphasize, on the other hand, that the item with more significant differences in each one of the analysed territorial contexts is the one that makes reference to the problems of desertification in the Iberian Peninsula originated by CC, but, in addition, within the own territorial contexts, the valuations on this problem are very disparate, what makes think that, these assessments are not due to the different rainfall levels that characterise the three territorial contexts, but that, possibly, the fact of belonging to different socio-cultural contexts also generates different self-perceptions of information with regard to causes, consequences, measures of struggle and pro-environmental attitudes.

Therefore, a social representation similar to the rest of the previously mentioned studies is being created, since it can be verified that when analysing the given answers, a similar pattern is followed in the three contexts and that, in addition, they coincide with the data of different study samples to this one.

On the other hand, it can also be observed that through the answers given by the students surveyed, although they are generally considered to have defined pro-environmental attitudes, this does not influence the social representation of CC either, but rather, the socio-cultural context.

In summary, the most significant differences found in the comparison of the three territorial contexts have been, on the one hand, that the students of the $TC_{\#3}$ are those who perceive themselves to be less individually responsible for the causes of CC as well as their country. On the other hand, in relation to the consequences of CC, once again the $TC_{\#3}$ is the one that perceives itself best informed in spite of obtaining the lowest score in this category, being, specifically, those that do not value as an effect of CC that reduces rainfall in their country of residence.

Finally, the $TC_{\#3}$ continues to be the one who feels best informed about solutions to fight against the CC. However, when assessing the knowledge about causes, consequences and solutions of the CC, a more or less similar pattern of response is generated, so the difference lies in the self-perception of the information and not really in the knowledge or social representation they have about the CC.

## 8. Conclusions

To conclude, it can be concluded that university students in the three climatological contexts analysed have a fairly high degree of climate literacy on the relationship between water issues and the climate crisis (extreme weather events, rising sea levels, desertification, etc.) and understand the close relationship between them. However, one might ask why the students of the Territorial Context 3 think that CC will not reduce rainfall in their country when, however, they think that the problems of desertification in the Iberian Peninsula will worsen ($O_{\#1}$).

On the other hand, the climate literacy of the students corresponds very much with the pro-environmental attitudes they express and with the information they claim to have on different aspects of climate change and its relationship with water, but this correspondence is not evident in aspects related to the biophysical processes related to CC, generating great confusion that may generate social representations of CC that may lead to mistaken ideas that may interfere with coherent pro-environmental decision-making at the individual level ($O_{\#2}$).

Finally, when comparing the results in the three contexts analysed, it is concluded that territory or climate do not favour important differences in the social representation of university students in the relations between water and CC, but the results do show that it is the common globalised culture around this phenomenon that generates a social representation that coincides in many aspects with that of people from other countries, of different ages and academic degrees ($O_{\#3}$).

Perhaps future lines of research should be aimed at understanding other types of university contexts with broader treatments of the climate crisis in their curricula, as has been done on other occasions in primary schools [76], which increase the level of climate literacy of students, to see if this training intensity significantly affects positive responses that help mitigate CC. It would also be interesting to know contexts where there is broad media coverage and quality on the causes, consequences and solutions of CC to see if it propitiates the generation of more climate literate individuals in all aspects and, therefore, with greater pro-environmental attitudes. On the other hand,

and according to the results obtained, it would be interesting to urge the competent authorities to make reforms in the curricula of all the formative stages, including in these specific topics where the interactions generated by the processes of human development with the environment can be explored and understood, and therefore, the consequences that this has in the aggravation of global warming. Trained people with the capacity to analyse and criticize the information and training they receive, will be people who can make wise decisions consistent with their lifestyles and consumption habits that do not interfere in the natural processes of the planet.

**Author Contributions:** Conceptualization, A.E.-R., J.G.-P. and P.Á.M.-C.; methodology, A.E.-R., J.G.-P. and P.Á.M.-C.; software A.E.-R., J.G.-P.; validation, A.E.-R., J.G.-P. and P.Á.M.-C.; formal analysis, A.E.-R. and J.G.-P.; investigation, A.E.-R., J.G.-P. and P.Á.M.-C.; resources, A.E.-R. and P.Á.M.-C.; data curation, A.E.-R., J.G.-P. and P.Á.M.-C.; writing—original draft preparation, A.E.-R. and J.G.-P.; writing—review and editing, A.E-R, J.G.-P. and P.Á.M.-C.; visualization, A.E.-R.; supervision, P.Á.M.-C. and J.G.-P.; project administration, P.Á.M.-C.; funding acquisition, P.Á.M.-C. and J.G.-P. All authors have read and agreed to the published version of the manuscript.

**Funding:** This research is part of the Resclima-Edu2 Project, "Educacion para el cambio climatico en educacion secundaria: investigacion aplicada sobre representaciones y estrategias pedagogicas en la transicion ecologica" funded by the Ministry of Economy and Competitiveness of the Government of Spain and the European Regional Development Fund (ERDF), Ref. RTI2018-094074-B-100; and the "Sustainability in Higher Education: Evaluation of the scope of the 2030 Agenda in curriculum innovation and teacher professional development in Andalusian Universities", Ref. B-SEJ-424-UGR18.

**Conflicts of Interest:** Authors do not declare conflicts of interest.

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
