# Peer review of "Water and Climate Change, Two Key Objectives in the Agenda 2030: Assessment of Climate Literacy Levels and Social Representations in Academics from Three Climate Contexts"

_water, doi:10.3390/w12010092_

Round 1

Reviewer 1 Report

It is very interesting and useful. Please check the attached comments.

Author Response

Dear reviewer,

Thank you very much for rating the article and for recommending it for publication after minor changes.

Each and every one of the comments have been taken care of carefully and properly as they have brought clarity and better structure to the text.

Next, we indicate the changes made in it:

Abstract: It is too simple. Please add some interesting findings from the assessment of climate literacy levels and social representations in academics from three climate contexts because you have obtained many good findings.

The most outstanding results have been added in a general way.

In the 1-3 Sections, some sentences need references to fully express your topics related water sustainable development, and water-related events:

Duan, W., Chen, Y., Zou, S. and Nover, D., 2019. Managing the water-climate-food nexus for sustainable

Zou, S., Jilili, A., Duan, W., Maeyer, P.D. and de Voorde, T.V., 2019. Human and Natural Impacts on the Water Resources in the Syr Darya River Basin, Central Asia. Sustainability, 11(11), p.3084.

Some ideas referring to these articles have been added to the text in section 2, as they provide some more examples on the subject being dealt with in the work.

Please write some words to link the Section 1-2 between Section 3.

A sentence linking section 1-2 to section 3 has been added at the end of section 2.

Tables 1-3: Please give the time period for the mean values.

Information is added on the time period to which the data in the tables refer.

Se añade información del periodo de tiempo al que hacen referencia los datos de las tablas.

Figures 2-4: It is strange to show in this way. Maybe you should use sub-figures. Also, please enlarge the font because it is too small.

The figures have been changed for others from a more scientific source (World Meteorological Organization).

The sizes of the figures have been changed.

Section 3.2: is it possible to give a table to summarize these paragraphs?

Thanks, we agree with the reviewer in a table that would allow us to have a global and synthetic view of the information contained in the heading. Although if we add more information to the manuscript it will be a more extensive text. We think it is better not to increase the number of pages, because it is already extensive.

Lines 409-411: please explain more clearly.

H3 has been reformulated to be better understood.

Figures in Results: please modify the sequences. Should be from Figure 5?

The sequences of the figures have been modified in the figures themselves and in the text.

Please give more sentences to discuss the uncertainty about your results.

A paragraph is added as a summary to highlight the most significant differences found in the study.

It is very interesting and useful. What we can do according to the results?

A final paragraph has been included in the conclusions that mentions what could be done according to the results obtained.

Reviewer 2 Report

The general assessment is that the paper is very good and that it gives a specific view into the problems associated with different climate change aspects in the context of perceptions held by the university student population in the three selected regions of the Iberian Peninsula, and it can almost be accepted in the proposed form. However, several minor suggestions are given below to have the paper presented even more clearly.

Ref. Abstract – The same holds true for Chapter 8. Conclusions – Put greater emphasis on the differences in the results obtained from the questionnaire in three different territorial contexts by individual questions.

Ref. 2.1. In addition to the general presentation of different manifestations of water crisis on the level of the planet Earth and its individual regions, mostly distant from the studied area, it would be good to refer to the results associated with the problem in the studied area itself or in the wider Mediterranean region, since even such regional analyses have an important impact on CC perception. Related to the problem of high water events, it is presented in Chapter 9, but there are no regional examples related to the problem of intensifying periods of drought on the Iberian Peninsula and the associated water use issues.

Ref. 3.1. Figure 1 is too small; together with its expansion, it would definitely be good to mark the three regions covered by the study in question.

Ref. Table 1-Table 3. In the description of the studied area, the rainfall is measured using mm/m3 as a unit of measurement, which is not appropriate. It is necessary to use either only mm without relating it to the volume unit or L/m2.

Figures 2, 3, 4 were taken over from a website, resulting in the problem that each figure has its own altitude measurement scale, with the y axis having different scales of the recorded monthly rainfall and temperatures. It is necessary to harmonize the measurement scales and generate new figures.

Ref 5.3. Please reference the tests used.

Ref. 6. The presentation of results starts suddenly and in detail. Some sort of an introductory sentence/chapter is missing to introduce the manner in which the results will be presented.

Ref. 7. In the Discussion, put a greater emphasis on what has been identified as differing significantly in different studied areas. Equally, point out the joint features.

Author Response

Dear reviewer,

Thank you very much for rating the article and for recommending it for publication after minor changes.

Each and every one of the comments have been taken care of carefully and properly as they have brought clarity and better structure to the text.

Next, we indicate the changes made in it:

Ref. Abstract – The same holds true for Chapter 8. Conclusions – Put greater emphasis on the differences in the results obtained from the questionnaire in three different territorial contexts by individual questions.

The most outstanding results have been generally added according to the categories studied and more emphasis has been placed on the differences found in the contexts studied by individual categories at the end of chapter 7 (discussion), thus leaving chapter 8 (conclusions) more as a reflection on the subject addressed and on what can be done with the results obtained.

Ref. 2.1. In addition to the general presentation of different manifestations of water crisis on the level of the planet Earth and its individual regions, mostly distant from the studied area, it would be good to refer to the results associated with the problem in the studied area itself or in the wider Mediterranean region, since even such regional analyses have an important impact on CC perception. Related to the problem of high water events, it is presented in Chapter 9, but there are no regional examples related to the problem of intensifying periods of drought on the Iberian Peninsula and the associated water use issues.

New information has been included on the specific problems suffered by the Iberian Peninsula in relation to the relationship between the water crisis and climate change. However, we thought that this information should be introduced in section 3 (background) in order to better contextualize the territories we have studied.

Ref. 3.1. Figure 1 is too small; together with its expansion, it would definitely be good to mark the three regions covered by the study in question.

The image has been expanded and the territorial contexts studied have been added to place them on the map.

Ref. Table 1-Table 3. In the description of the studied area, the rainfall is measured using mm/m3as a unit of measurement, which is not appropriate. It is necessary to use either only mm without relating it to the volume unit or L/m2.

The unit of measurement is changed to mm. The data do not change as a typographical error has been made previously.

Figures 2, 3, 4 were taken over from a website, resulting in the problem that each figure has its own altitude measurement scale, with the y axis having different scales of the recorded monthly rainfall and temperatures. It is necessary to harmonize the measurement scales and generate new figures.

The figures have been changed for others from a more scientific bibliographic source (World Meteorological Organization). The time ranges shown by the data in the figures are also added.

Ref 5.3. Please reference the tests used.

The text indicates the tests carried out, but it has been added with which statistical software they were made.

Ref. 6. The presentation of results starts suddenly and in detail. Some sort of an introductory sentence/chapter is missing to introduce the manner in which the results will be presented.

A new paragraph has been added to introduce the way in which results are presented.

Ref. 7. In the Discussion, put a greater emphasis on what has been identified as differing significantly in different studied areas. Equally, point out the joint features.

A paragraph is added as a summary to highlight the most significant differences found in the study.